# Interpreting wide-band neural activity using convolutional neural networks

Markus Frey[1,2]*, Sander Tanni[3], Catherine Perrodin[4], Alice O'Leary[3], Matthias Nau[1,2], Jack Kelly[5], Andrea Banino[6], Daniel Bendor[4], Julie Lefort[3], Christian F Doeller[1,2,7]†, Caswell Barry[3]†*

[1]Kavli Institute for Systems Neuroscience, Centre for Neural Computation, The Egil and Pauline Braathen and Fred Kavli Centre for Cortical Microcircuits, NTNU, Norwegian University of Science and Technology, Trondheim, Norway; [2]Max-Planck-Insitute for Human Cognitive and Brain Sciences, Leipzig, Germany; [3]Cell & Developmental Biology, UCL, London, United Kingdom; [4]Institute of Behavioural Neuroscience, UCL, London, United Kingdom; [5]Open Climate Fix, London, United Kingdom; [6]DeepMind, London, United Kingdom; [7]Institute of Psychology, Leipzig University, Leipzig, Germany

*For correspondence:
markus.frey@ntnu.no (MF);
caswell.barry@ucl.ac.uk (CB)

†These authors contributed equally to this work

Competing interests: The authors declare that no competing interests exist.

**Abstract** Rapid progress in technologies such as calcium imaging and electrophysiology has seen a dramatic increase in the size and extent of neural recordings. Even so, interpretation of this data requires considerable knowledge about the nature of the representation and often depends on manual operations. Decoding provides a means to infer the information content of such recordings but typically requires highly processed data and prior knowledge of the encoding scheme. Here, we developed a deep-learning framework able to decode sensory and behavioral variables directly from wide-band neural data. The network requires little user input and generalizes across stimuli, behaviors, brain regions, and recording techniques. Once trained, it can be analyzed to determine elements of the neural code that are informative about a given variable. We validated this approach using electrophysiological and calcium-imaging data from rodent auditory cortex and hippocampus as well as human electrocorticography (ECoG) data. We show successful decoding of finger movement, auditory stimuli, and spatial behaviors – including a novel representation of head direction - from raw neural activity.

## Introduction

A central aim of neuroscience is deciphering the neural code, understanding the neural representation of sensory features and behaviors, as well as the computations that link them. The task is complex, and although there have been notable successes – such as the identification of orientation selectivity in V1 (*Hubel et al., 1959*) and the representation of self-location provided by hippocampal place cells (*O'Keefe and Dostrovsky, 1971*) – progress has been slow. Neural activity is high dimensional and often sparse, while the available datasets are typically incomplete, being both temporally and spatially limited. This problem is compounded by the fact that the code is multiplexed and functionally distributed (*Walker et al., 2011*). As such, activity in a single region may simultaneously represent multiple variables, to differing extents, across different elements of the neural population. Taking the entorhinal cortex for example, a typical electrophysiological recording might contain spike trains from distinct cells predominantly encoding head direction, self-location, and movement speed via their firing rates (*Sargolini et al., 2006*; *Kropff et al., 2015*; *Hafting et al., 2005*), while other neurons have more complex composite representations (*Hardcastle et al., 2017*). At the same time, information about speed and location can also be identified from the local field potential (LFP) (*McFarland et al., 1975*) and the relative timing of action potentials

(*O'Keefe and Recce, 1993*). Fundamentally, although behavioral states and sensory stimuli can generally be considered to be low dimensional, finding the mapping between noisy neural representations and these less complex phenomena is far from trivial.

Historically, the approach for identifying the correspondence between neural data and external observable states – stimuli or behavior – has been one of raw discovery. An experimenter, guided by existing knowledge, must recognize the fact that the activity covaries with some other factor. Necessarily this is an incremental process, favoring identification of the simplest and most robust representations, such as the sparse firing fields of place cells (*Muller et al., 1987*). Classical methods, like linear regression and linear-nonlinear-Poisson cascade models (*Corrado et al., 2005*; *Kropff et al., 2015*), provide powerful tools for the characterization of existing representations but are less useful for the identification of novel responses – they typically require highly processed data in conjunction with strong assumptions about the neural response, and in the former cases are limited to one dimensional variables. Recent advances in machine learning suggest an alternative strategy. Artificial neural networks (ANNs) trained using error backpropagation regularly exceed human-level performance on tasks in which high dimensional data is mapped to lower dimensional labels (*Krizhevsky et al., 2012*; *Mnih et al., 2015*). Indeed, these tools have successfully been applied to *processed* neural data – accurately decoding behavioral variables from observed neural firing rates (*Glaser et al., 2017*; *Tampuu et al., 2018*). However, the true advantage of ANNs is not their impressive accuracy but rather the fact that they make few assumptions about the structure of the input data and, once trained, can be analyzed to determine which elements of the input, or indeed combination of elements, are most informative (*Cichy et al., 2019*; *Hasson et al., 2020*; *Cammarata et al., 2020*). Moreover, this framework provides full control over the weights, activations, and objective functions of the model, allowing fine-grained analysis of the inner workings of the network. Viewed in this way ANNs potentially provide a means to accelerate the discovery of novel neural representations.

To test this proposal, we developed a convolutional network (*LeCun et al., 2015*) able to take minimally processed, wide-band neural data as input and predict behaviors or other co-recorded stimuli. In the first instance, we trained the model with unfiltered and unclustered electrophysiological recordings made from the CA1 pyramidal cell layer in freely foraging rodents. As expected, the network accurately decoded the animals' location, speed, and head direction – without spike sorting or additional user input. Analysis of the trained network showed that it had 'discovered' place cells (*O'Keefe and Dostrovsky, 1971*; *O'keefe and Nadel, 1978*) – frequency bands associated with pyramidal waveforms being highly informative about self-location (*Epsztein et al., 2011*). Equally, it successfully recognized that theta-band oscillations in the LFP were informative about running speed (*McFarland et al., 1975*; *Jeewajee et al., 2008*). Unexpectedly, the network also identified a population of putative CA1 interneurons that encoded information about head direction. We corroborated this observation using conventional tools, confirming that the firing rate of these neurons was modulated by facing-direction, a previously unreported relationship. Beyond this we found the trained network provided a means to efficiently conduct analyses which would otherwise have been complex or time consuming. For example, comparison of all frequency bands revealed positive interactions between frequencies associated with waveforms – components of the neural code that convey more information together than when considered individually. Subsequently, to demonstrate the generality of this approach, we applied the same architecture to electrophysiological data from auditory cortex, two-photon calcium imaging data acquired while mice explored a virtual environment as well as ECoG recordings in humans performing finger movements (*Schalk et al., 2007*).

Our model differs markedly from conventional decoding methods which often use Bayesian estimators (*Zhang et al., 1998*) for hippocampal recordings in conjunction with highly processed neural data or linear methods for movement decoding (*Antelis et al., 2013*). In the case of extracellular recordings, this usually implies that time-series are filtered and processed to detect action potentials and assign them to specific neurons. Necessarily this discards information in frequency bands outside of the spike range, potentially introducing biases implicit in the algorithm used (*Pachitariu et al., 2016*; *Chung et al., 2017*; *Hyung et al., 2017*) and operator's subjective preferences (*Harris et al., 2000*; *Wood et al., 2004*), and – despite considerable advances – still demands manual input to adjust clusters (*Pachitariu et al., 2016*). Furthermore, accurate calculation of prior expectations regarding the way in which the data varies with the decoded variable – an essential component of Bayesian decoding – requires considerable knowledge about the structure of the

neural signal being studied and appropriate noise models. Other authors have attempted to address some of these shortcomings, for example, decoding without assigning action potentials to specific neurons (*Kloosterman et al., 2014*; *Ackermann et al., 2019*; *Deng et al., 2015*) or combining LFP and spiking data (*Stavisky et al., 2015*) for cursor control in patients. However, these approaches relied on existing assumptions about neural coding statistics and did not use all available information in the recordings, while their primary focus was simply to improve decoding accuracy. In contrast, the flexible, general-purpose approach we describe here requires few assumptions and – once trained – can be interrogated to inform the discovery of novel neural representations. In addition, as the model does not rely on specific oscillations or spike waveforms, it can easily generalize across domains – a fact we demonstrate with optical imaging data acquired in rodents as well as human ECoG signals.

## Results

In the following section, we present our model, the results, and describe how it was applied across different datasets. First, we report results obtained when decoding spatial behaviors from CA1 recordings made in freely moving rats. Second, we describe how the model results are interpreted and which informative features drive decoding performance, including a more detailed analysis of head direction and speed decoding. Third, we show that the model generalizes to different brain regions and recording techniques, including calcium imaging in mice and ECoG recordings in humans. In short, the framework provides a unified way of decoding continuous behaviors or stimuli from neural time-series data. The model first transforms the neural data into frequency space via a wavelet transformation. Next, the outputs (behaviors or stimuli) are resampled to match the sample rate of the neural data. Finally, the convolutional neural network takes the transformed neural data as input and decodes each output separately.

### Accurate decoding of self-location from CA1 recordings

In the first instance, we sought to evaluate our network-based decoding approach on well character-ised neural data with a clear behavioral correlate. To this end, we used as input extracellular electro-physiological signals recorded from hippocampal region CA1 in five freely moving rats – place cells from this area being noted for their spatially constrained firing fields that convey considerable infor-mation about an animal's self-location (*O'Keefe and Dostrovsky, 1971*; *Muller et al., 1987*). Ani-mals were bilaterally implanted with 32 tetrodes and, after recovery and screening, 128 channel wide-band (0 Hz to 15000 Hz sampled at 30 kHz) recordings were made while the rats foraged in a 1.25 x 1.75 m arena for approximately 40 min (see methods). Raw electrophysiological data were decomposed using Morlet wavelets to generate a three-dimensional representation depicting time, channels, and frequencies from 2 Hz to 15,000 Hz (*Figure 1A*; *Christopher and Gilbert, 1998*). Using the wavelet coefficients as inputs, the model was trained in a supervised fashion using error backpropagation with the X and Y coordinates of the animal as regression targets. We report test-set performance for fully cross-validated models using five splits across the whole duration of the experiment.

To reduce computational load and improve test set generalization, we use 2D-convolutions with shared weights applied to the three-dimensional input (*Figure 1B*, *Supplementary file 1*) – the first eight convolutional layers having weights shared across channels and the final six across time. Imple-menting weight sharing in this way is desirable as the model is able to efficiently identify features that reoccur across time and channels, for example, prominent oscillations or waveforms, while also drastically reducing model complexity. For comparison, an equivalent architecture trained to decode position from 128 channels of hippocampal electrophysiological but without shared weights had 38,144,900 hyperparameters compared to 5,299,653 – an increase of 720%. The more complex model took 4.7 hr to run per epoch, as opposed to 175 s, and ultimately yielded less accurate decoding (*Figure 1—figure supplement 1*).

The model accurately decoded position from the unprocessed neural data in all rats, providing a continuous estimate of location with a mean error less than 10% of the environment's length. This demonstrates that, as expected, the network was able to identify informative signals in the raw neu-ral data (Mean error 17.31 cm ± 4.46 cm; Median error 11.40 cm ± 3.82 cm; Chance level 65.03 cm ± 6.91 cm; *Figure 1C,D*). To provide a familiar benchmark, we applied a standard Bayesian decoder

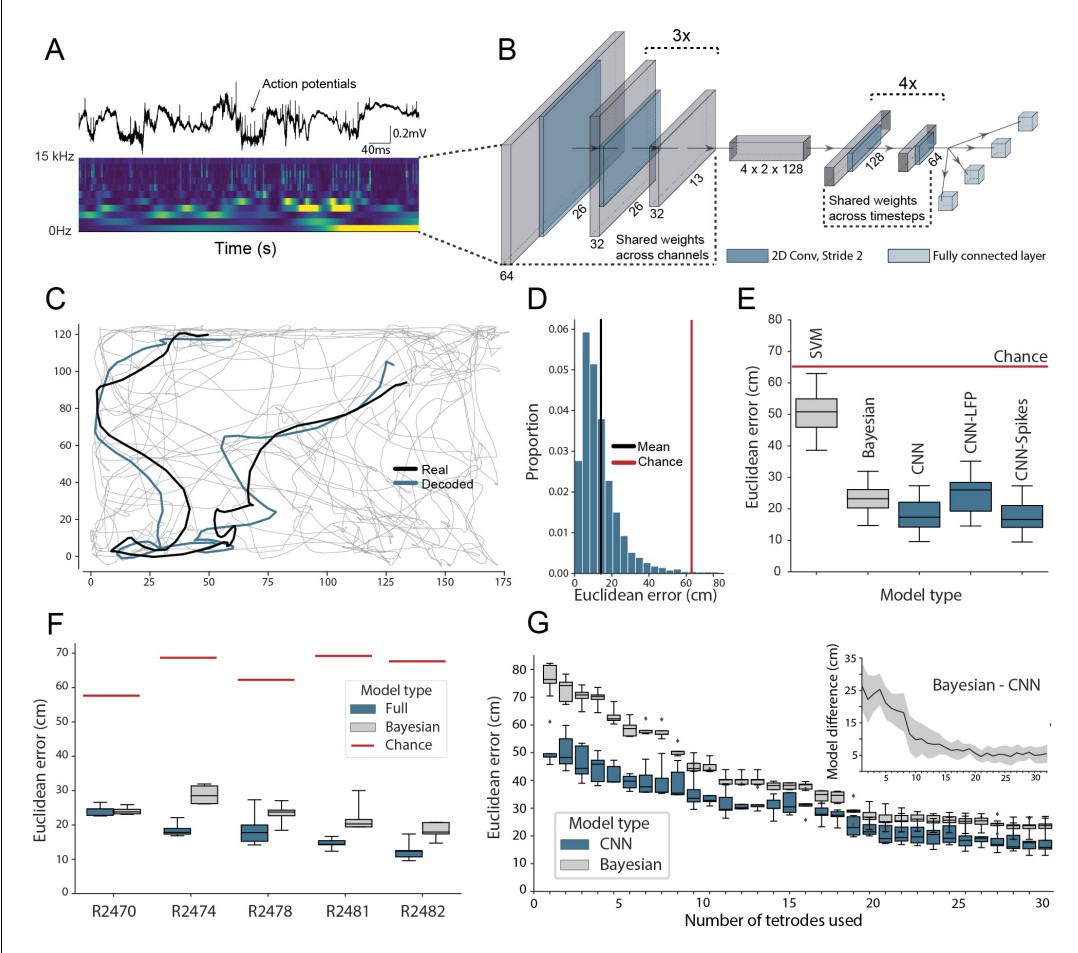

**Figure 1.** Accurate decoding of self-location from unprocessed hippocampal recordings. (A) Top, a typical 'raw' extracellular recording from a single CA1 electrode. Bottom, wavelet decomposition of the same data, power shown for frequency bands from 2 Hz to 15 kHz (bottom to top row). (B) At each timestep wavelet coefficients (64 time points, 26 frequency bands, 128 channels) were fed to a deep network consisting of 2D convolutional layers with shared weights, followed by a fully connected layer with a regression head to decode self-location; schematic of architecture shown. (C) Example trajectory from R2478, true position (black) and decoded position (blue) shown for 3 s of data. Full test-set shown in *Figure 1—video 1*. (D) Distribution of decoding errors from trial shown in (C), mean error (14.2 cm ± 12.9 cm, black), chance decoding of self-location from shuffled data (62.2 cm ± 9.09 cm, red). (E) Across all five rats, the network (CNN) was more accurate than a machine learning baseline (SVM) and a Bayesian decoder (Bayesian) trained on action potentials. This was also true when the network was limited to high-frequency components (>250Hz, CNN-Spikes). When only local frequencies were used (<250Hz, CNN-LFP), network performance dropped to the level of the Bayesian decoder (distributions show the fivefold cross validated performance across each of five animals, n=25). Note that this likely reflects excitatory spikes being picked up at frequencies between 150 and 250 Hz (*Figure 1—figure supplement 2*). (F) Decoding accuracy for individual animals, the network outperformed the Bayesian decoder in all cases. An overview of the performance of all tested models can be seen in *Figure 1—figure supplement 3*. (G) The advantage of the network over the Bayesian decoder increased when the available data was reduced by downsampling the number of channels (data from R2478). Inset shows the difference between the two methods.

The online version of this article includes the following video and figure supplement(s) for figure 1:

**Figure supplement 1.** Effect of weight sharing on model performance.
**Figure supplement 2.** Decoding performance of LFP models.
**Figure supplement 3.** Decoding performance across different models.
**Figure supplement 4.** Decoding performance across time windows and continuity priors.
**Figure supplement 5.** Difference in error distribution between Bayesian decoder and CNN.
**Figure supplement 6.** Decoding performance with different speed thresholds.
**Figure supplement 7.** Detection of replay events in CA1 recordings.
**Figure supplement 8.** Influence of decoding across channels.
**Figure supplement 9.** A subset of frequency pairs exhibit greater than expected decoding influence.
**Figure supplement 10.** Effect of downsampling on model performance.
**Figure 1—video 1.** Decoding of multiple behaviors from rodent CA1.

*Figure 1 continued on next page*

without a continuity prior (*Ólafsdóttir et al., 2015*) to the spiking data from the same datasets (see Materials and methods, see also *Figure 1—figure supplement 4B* for comparison to Bayesian decoder with continuity prior; *Zhang et al., 1998*). To this end, action potentials were identified, clustered, manually curated, and spike time vectors were used to decode location – data contained in the local field potential (LFP) was discarded. Note that while the Bayesian decoder explicitly incorporates information about the probability with which animals visit each spatial location, the CNN is effectively feed-forward – being presented with ~2 s windows of data. Our model was consistently more accurate than the Bayesian decoder, exceeding its performance in all animals (Bayesian mean error 23.38 cm ± 4.35 cm; network error 17.31 cm ± 4.46 cm; Wilcoxon signed-rank test: T=18, p=0.0001). The CNN's advantage over the Bayesian decoder was in part derived from the fact that it made fewer large errors (*Figure 1—figure supplement 5*), thus the median errors for the two methods were more similar (all channels; Bayesian decoder median error 13.99 cm ± 1.77 cm, network median error 11.40 cm ± 3.82 cm). The high accuracy and efficiency of the model for these harder samples suggest that the CNN utilizes additional information from sub-threshold spikes and those that were not successfully clustered and/or nonlinear information which is not available to the Bayesian decoder.

The relative advantage over the Bayesian decoder increased further when the number of channels used for decoding was downsampled to simulate smaller recordings (linear regression Wald-test (n=31), s=-0.65, p=1.83e-10; *Figure 1G*). Notably, the model achieved a similar mean decoding performance with twenty tetrodes (80 channels, 23.45 cm ± 3.15 cm) as the Bayesian decoder reached with the full data set (128 channels, 23.25 cm ± 2.79 cm, *Figure 1G*). Note that for these comparisons, the Bayesian decoder was only applied to periods when the animal was travelling at >3cm/s, in contrast the CNN did not have a speed threshold (i.e. is trained on moving and stationary periods). To confirm this difference did not favor the CNN, we examined how its performance compared to the Bayesian decoder if both were subject to a range of speed thresholds. As expected, limiting the CNN to only periods of movement actually improves its performance, accentuating the difference between it and the Bayes decoder (0 cm/s speed threshold: Bayesian mean error 22.23 cm ± 3.72 cm; network error 17.37 cm ± 3.58 cm; 25 cm/s speed threshold: Bayesian mean error 17.17 cm ± 2.788 cm; network error 13.40 cm ± 3.59 cm, *Figure 1—figure supplement 6*). To compare the CNN against standard machine learning tools, we used the wavelet transformed data to train support vector machines (SVMs). Note that in this case, the spatial structure of the input is inevitably lost as the input features are transformed to a one-dimensional representation. Both linear (53.6 cm ± 14.77 cm; *Figure 1E*) and non-linear SVMs (61.2 cm ± 15.67 cm) performed considerably worse than the CNN. Note that it is computationally not feasible to use a fully connected neural network as the flattening of the spatial structure will inevitably lead to a multi-million parameter model. These models are notoriously hard to train and will not fit many consumer-grade GPUs, in contrast to convolutional neural networks which share the weights across the spatial dimension.

To better understand which elements of the raw neural data the network used, we retrained our model using datasets limited to just the LFP (<250Hz) and just the spiking data (>250Hz). In both cases, the network accurately decoded location (spikes-only (CNN-Spikes) mean error 17.23 cm ± 4.69 cm; LFP-only (CNN-LFP) mean error 24.24 cm ± 6.00 cm; *Figure 1E*), indicating that this framework is able to extract information from varied electrophysiological sources. Consistent with the higher information content of action potentials, the spikes-only network was considerably more accurate than the LFP-only network (Wilcoxon signed-rank test, two-sided (n=25): T=0, p=1.22e-05), although the LFP-only network was still comparable with the spike-based Bayesian decoder (Bayesian 23.38 cm ± 4.35 cm; LFP 24.24 cm ± 6.00 cm; Wilcoxon signed-rank test, two-sided (n=25): T=136, p=0.475). If we retrain the model on frequency bands 0–150 Hz and 150–250 Hz, we observe that spatial information is predominantly contained in higher band frequencies (*Figure 1—figure supplement 2*, see also *Figure 3—figure supplement 1*), likely reflecting power from pyramidal cell waveforms reaching these frequencies. Note that previous studies have shown that demodulated theta is informative about the position of an animal in its environment (*Agarwal et al., 2014*).

However, in those experiments theta oscillations were converted into a complex-valued signal, which carried both the magnitude and phase of theta – here we only used the magnitude for decoding of position.

The standard decoding model downsamples the wavelet frequencies to a rate of 30 Hz, potentially discarding transient non-local representations (e.g. replay events and theta sequences). To evaluate the potential of our model to detect these short-lived events, we applied a lower downsampling factor to achieve a sampling rate of 500 Hz. This allowed us to investigate decoded time points during immobility periods in which ripples were detected in the LFP data using standard methods (see Materials and methods). Examining these periods, we saw the CNN often decoded transient, high velocity trajectories that resembled those reported in studies of open field replay which are known to co-occur with ripples (see *Figure 1—figure supplement 7A*). Consistent with this interpretation the decoding error for these periods (i.e. Euclidean distance between the animal's location and decoded location) was larger than for matched periods when the animal was stationary but ripples were not present (n=1000, p<0.001, *Figure 1—figure supplement 7B*). We furthermore found these putative replay trajectories were longer than trajectories decoded from the matched stopped periods (n=1000, p=0.003, *Figure 1—figure supplement 7C*) but were coherent, similar to trajectories decoded during movement (n=1000, p=0.257, *Figure 1—figure supplement 7D*). Taken together, these analyses indicate that our model captures transient high-velocity trajectories occurring during sharp-wave ripples but not during other stationary periods – it is highly likely that these correspond to replay events. Thus, it seems plausible that non-local representations are accessible to this CNN framework.

## Simultaneous decoding of multiple factors

The hippocampal representation of self-location is arguably one of the most readily identifiable neural codes – at any instance a small number of sparsely active neurons are highly informative. To provide a more stringent test of the network's ability to detect and decode behavioral variables from unprocessed neural signals, we retrained with the same data but simultaneously decoded position, speed, and head-direction within a single model. CA1 recordings are known to incorporate information about these additional factors but their representation is less pronounced than that for self-location. Thus, the spatial activity of place cells is known to be weakly modulated by head direction (*Jercog et al., 2019*; *Yoganarasimha et al., 2006*), while place cell firing rates and both the frequency and amplitude of theta, a 7–10 Hz LFP oscillation, are modulated by running speed (*McFarland et al., 1975*; *Jeewajee et al., 2008*). In this more complex scenario, the architecture and hyper-parameters remained the same with just the final fully connected layer of the network being replicated, one layer for each variable, with the provision of appropriate loss functions – cyclical mean absolute error for head direction and mean absolute error for speed (see Materials and methods). All three variables were decoded simultaneously and accurately (Position, 17.78 cm ± 4.96 cm; Head Direction, 0.80 rad ± 0.18 rad; Speed 4.94 cm/s ± 1.00 cm/s; *Figure 2A* and *Figure 1—video 1*), with no meaningful decrement in performance relative to the simpler network decoding only position (position-only model 17.31 cm ± 4.46; combined model 17.78 cm ± 4.96 cm; Wilcoxon signed-rank test two-sided (n=25): T=116, p=0.2108). Note that, a Bayesian decoder trained to decode head direction achieves a performance of 0.97 rad ± 0.14 rad using spike sorted neural data, significantly worse than our model (Wilcoxon signed-rank test two-sided (n=25): T=47, p=0.0018) but more accurate than would be expected by chance (Wilcoxon signed-rank test two-sided (n=25): T=0, p=1.22e-5). The $R^2$-score metric from the fully trained network – a measure which represents the portion of variance explained and is independent of the loss function – indicated that mean decoding performance was above chance for all three behaviors ($R^2$-score Position 0.86 ± 0.08, Head Direction 0.60 ± 0.12, Speed 0.72 ± 0.14, Chance $R^2$-score Position −0.14 ± 0.13, Head Direction 0.04 ± 0.11, Speed −0.16 ± 0.22) (*Figure 2B*). Thus, the network was able to effectively access multiplexed information embedded in minimally processed neural data.

## Interrogation of electrophysiological recordings

Although the network supports accurate decoding of self-location from electrophysiological data, this was not our main aim. Indeed, our primary goal for this framework was to provide a flexible tool capable of discovering and characterising sensory and behavioral variables represented in neural

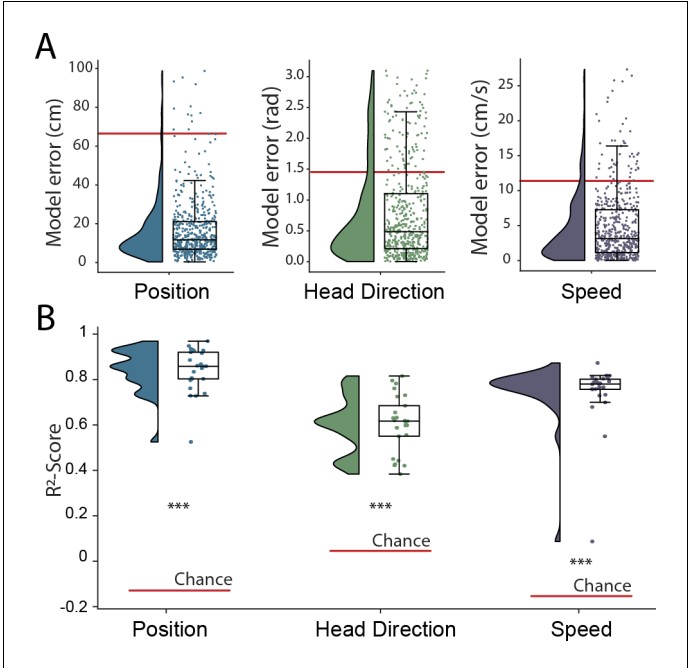

**Figure 2.** Simultaneous decoding of multiple variables from hippocampal data. (**A**) Position, head direction, and running speed were accurately decoded in concert by a single network. Data from all five animals, each point indicates an error for a single sample. The red dashed line indicates the chance level obtained by shuffling the input relative to the output while fully retraining the model. (**B**) $R^2$-scores, a loss-invariant measure of model performance – ranging from 1 (perfect decoding) to negative infinity – allowing performance to be compared between dissimilar variables. Data as in (**A**), each point corresponds to one of five cross-validations within each of five rats.

The online version of this article includes the following figure supplement(s) for figure 2:

**Figure supplement 1.** Running speed linearly correlates with power in multiple frequency bands.

---

data - providing insight about the form and content of encoded information. To this end, in the fully trained network, we used a shuffling procedure to estimate the influence that each element of the 3D input (frequencies x channel x time) had on the accuracy of the decoded variables (see Materials and methods). Since this approach does not require retraining, it provides a rapid and computationally efficient means of assessing the contribution made by different channels, frequency bands, and time points.

Turning first to position decoding, we saw that the adjacent 469 Hz and 663 Hz frequency bands were by far the most influential (*Figure 3A*). Since these recordings were made from CA1, we hypothesized that these frequencies corresponded to place cell action potentials. To confirm this hypothesis – and demonstrate that it was possible to objectively use this network-based approach to identify the neural basis of decoded signals – we applied the following approach (see Materials and methods): First, we isolated the waveforms of place cells (n=629) and putative interneurons (n=91) in all animals, which were identified using a conventional approach (*Pachitariu et al., 2016*; *Klausberger et al., 2003*; *Csicsvari et al., 1999*). Second, for these two groups, we calculated the relative representation of the 26 frequency bands in their waveforms. We found that the highly informative 469 Hz and 663 Hz bands were the dominant components of place cell action potentials and that in general the power spectra of these spikes strongly resembled the frequency influence plot for position decoding (Spearman rank-order correlation, two-sided (n=26) ρ=0.84, p=7.63e-08; *Figure 3B*). In contrast, putative interneurons – which typically have a shorter after-hyperpolarization than place cells (*English et al., 2017*) – were characterized by higher frequency components (*Figure 3B*, Mann-Whitney rank test interneuron (n=91) vs. place cell (n=629), U=1009.5, p=2.47e-13), with the highest power at 5304 Hz and 3750 Hz, bands that were considerably less informative about self-location (*Figure 3A*).

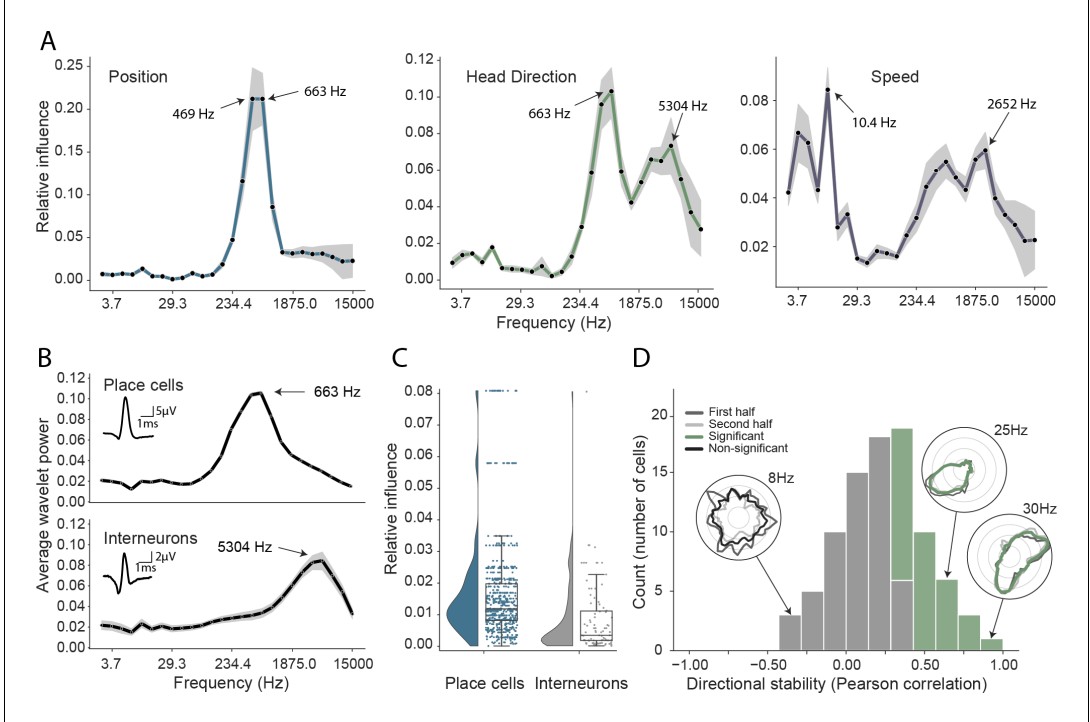

**Figure 3.** Analysis of trained network identifies informative elements of the neural code. (A) A shuffling procedure was used to determine the relative influence of different frequency bands in the network input. Left, the 469 Hz and 663 Hz components – corresponding to place cell action potentials – were highly informative about animals' positions. Middle, both place cells and putative interneurons (5304 Hz) carried information about head direction. Right, several frequency bands were informative about running speed, including those associated with the LFP (10.4 Hz) and action potentials. Data from all animals. (B) Wavelet coefficients of place cell (top) and interneuron (bottom) waveforms are distinct and correspond to frequencies identified in A. Inset, average waveforms. Data from all animals. (C) Frequency bands associated with place cells (469 and 663 Hz) were more informative about position than those associated with putative interneurons (5304 Hz) – their elimination produced a larger decrement in decoding performance (p<0.001). Data from all animals. (D) A subset of putative CA1 interneurons encodes head direction. Thirty-three of 91 interneurons from five animals exhibited pronounced directional modulation that was stable throughout the recording (green). Depth of modulation quantified using Kullback-Leibler divergence vs. uniform circle. Stability assessed with the Pearson correlation between polar ratemaps from the first and second half of each trial (dark gray and light gray). Cells with p<0.01 for both measures were considered to be reliably modulated by head direction. Inset, example polar ratemaps. Data from all animals.

The online version of this article includes the following figure supplement(s) for figure 3:

**Figure supplement 1.** Performance of standard model compared with models trained on single frequency bands.
**Figure supplement 2.** Decoding influence for downsampled models.
**Figure supplement 3.** Decoding influence for simulated behaviors.

Since the frequencies associated with place cell waveforms were the most informative, this indicated that the network had correctly identified place cells as the primary source of spatial information in these recordings. To corroborate this, we used the same data and for each channel eliminated power in the 469 Hz and 663 Hz frequency bands at time points corresponding to either place cell or interneuron action potentials. As expected, position decoding was most strongly affected by removal of the place cell time points (Mann-Whitney-U-Test (n=629 place cells, n=91 interneurons): U=1497, p=2.86e-08; *Figure 3C*). Using the same shuffling method, we also analyzed how informative each channel was about self-location (*Figure 1—figure supplement 8*). In particular, we found that the number of place cells identified on a tetrode from the spike sorted data was highly correlated with the tetrode's spatial influence (Spearman rank-order correlation (n=128) ρ=0.71, p=5.11e-06) and that the overall distribution of both number of place cells and spatial influence followed a log-normal distribution (Shapiro-Wilk test on log-transformed data, number of place cells, W=0.79, p=3.59e-05; tetrode influence W=0.59, p=3.04e-08; *Figure 1—figure supplement 8B*). In sum, this analysis correctly identified that the firing rates of both place cells and

putative interneurons are informative about an animal's location, place cells more so than interneurons (*Wilent and Nitz, 2007*). The analysis also highlighted the spatial activity of place cells, pointing to the stable place fields as a key source of spatial information.

A potential concern is that our approach might not identify multiple frequency bands if the information they contain is mutually redundant. The previous example, in which place cells and putative interneurons were both found to be informative about self-location, demonstrates this is not entirely the case. However, to further exclude this possibility, we compared the influential frequencies identified from our complete model with models trained on just a single frequency band at a time. Specifically, 26 models were trained, one for each frequency – the performance of each of these models being taken as an indication of the information present in that band. As expected we found both methods identified similar frequencies as indicated by a high correlation between our influence measure and the performance of single frequency band models (Position, Spearman rank-order correlation (n=26) ρ=0.88, p<0.001; Head Direction, ρ=0.82, p<0.001; Speed, ρ=0.47, p=0.02, *Figure 3—figure supplement 1*). Note that, although each model was individually faster to train than the complete model, the time to train all 26 was considerably longer than the single model applied simultaneously to all frequencies (51.2 hr vs. 8.6 hr, ~6x faster).

To further assess the robustness of the influence measure, we conducted two additional analyses. First, we varied the volume of data by recalculating the influence measure for different amounts of training data (1–35 min, see Materials and methods). We found that our model achieves an accurate representation of the true influence with as little as 5 min of training data (mean Pearson's r = 0.89 ± 0.06, *Figure 3—figure supplement 2*). Secondly, we assessed influence accuracy on a simulated behavior in which we varied the ground truth frequency information (see Materials and methods). The model trained on the simulated behavior is able to accurately represent the ground truth information (modulated frequencies 58 Hz and 3750 Hz, *Figure 3—figure supplement 3*).

Thus, our combined approach provides a fair and efficient means to determine the informative elements of wide-band neural data. More importantly, analyses of the full network enables multiple frequency bands to be considered in-concert, providing a means to identify interactions (e.g.*Figure 1—figure supplement 9*) that are not accessible to standard single-frequency methods.

## CA1 interneurons are modulated by head direction

Next, having validated our approach for spatial decoding, we examined the basis upon which the network was able to decode head direction. Although place cells primarily provide an allocentric spatial code, their infield firing rate is known to be modulated by heading direction (*Muller et al., 1994*; *Rubin et al., 2014*). Consistent with the presence of this directional code, we again saw that the most influential frequencies for head direction decoding were those associated with place cells (469 Hz and 663 Hz; *Figure 3A*). However, the distribution also incorporated a secondary peak corresponding to the frequencies typical for interneuron waveforms (Spearman rank-order correlation (n=26) ρ=0.76, p=5.71e-06, *Figure 3A&B*). Presubicular interneurons have been shown to be modulated by both head direction and angular velocity (*Preston-Ferrer et al., 2016*) but to the best of our knowledge no similar responses have been noted in CA1. To establish if putative interneurons conveyed information about head direction, we again used an 'elimination' analysis on data from all five animals – the two frequency bands most strongly associated with interneurons (3750 Hz and 5304 Hz) were scrambled at time points when interneuron spikes were present. Consistent with the influence plots, we found that selectively eliminating putative interneurons degraded the accuracy with which head direction was decoded (relative influence: 0.089 ± 0.043, two-sided t-test (n=91) t=4.16, p=0.014). As a final step, to verify this novel observation we reverted to a standard approach. Specifically, we calculated the directional ratemap for each interneuron using only periods when the animal was in motion (>10cm/s), determined the Kullback-Leibler divergence vs. a uniform circle (*Doeller et al., 2010*), and applied a shuffling procedure to determine significance – as a whole the population exhibited reliable but weak modulation of interneuronal firing rate by head direction (Kullback-Leibler Divergence (n=91): 0.0067 ± 0.009) with 58.2% (53/91) of cells being individually significant (p<0.01). Behaviors that are inhomogeneously distributed or confounded can result in spurious neural correlates (*Muller et al., 1994*). To control for this possibility, we repeated the analysis using only data from the centre of the environment (>25cm from the long sides of the enclosure and >20cm from the short sides). Additionally, to verify stability, we controlled that ratemaps generated from the first and second half of the trial were correlated (Pearson correlation,

p<0.01). Under this more rigorous analysis, we confirmed that a sub-population (36.2%, 33/91) of putative hippocampal interneurons were modulated by head direction, a previously unrecognized spatial correlate (*Figure 3D*).

## Multiple electrophysiological features contribute to the decoding of speed

The frequency influence plots for running speed also showed several local peaks (*Figure 3A*), that in all cases corresponded to established neural correlates. In rodents, theta frequency and power are well known to co-vary almost linearly with running speed (*McFarland et al., 1975*; *Jeewajee et al., 2008*), accordingly analysis of the network identified the 10.4 Hz frequency band as the most influential. Similarly, the firing rate of place cells increases with speed, an effect captured by the peak at 663 Hz. Interestingly a clear peak is also evident at 2652 Hz, indicating that interneuron firing rates are also informative – originating either from CA1 speed cells (*Góis and Tort, 2018*) or from theta-locked interneurons (*Huh et al., 2016*). Finally, a 4th peak was evident at 3.66 Hz and 5.17 Hz, a range that corresponds to type 2 ('atropine sensitive') theta which is present during immobility (*Kramis et al., 1975*; *Sainsbury et al., 1987*). To corroborate this conclusion, we calculated the correlation between power in each frequency band and running speed (*Figure 2—figure supplement 1A*), confirming that the latter band showed the expected negative correlation – higher power at low speeds – while the other three peaks were positively correlated.

## Generalization across brain regions and recording techniques

As a final step, we sought to determine how well our approach generalized to other recording techniques and brain areas. Addressing the latter point first, we trained the network using electrophysiological recordings (64 channels) from the primary auditory cortex of a freely moving mouse while pure tone auditory stimuli (4–64 kHz, duration 200 ms) were played from a speaker (*Figure 4A*). As above, the raw electrophysiological data was transformed to the frequency domain using Morlet wavelets and this wide-band frequency representation was used as input. The model architecture and hyperparameters were kept the same, reducing only the number of down-sampling steps because of the smaller input size (64 channels vs. 128 channels for CA1 recordings – each down-sampling layer halves the number of units in the previous layer). The auditory stimuli – training target – was modelled as a continuous variable with '−1' indicating no tone present and the log-transformed frequency of the sound at all other time points. As expected, this model architecture was also able to accurately decode tone stimuli from auditory cortex (Performance in original frequency space, $R^2$-score: 0.734 ± 0.080, chance model: −0.432 ± 0.682, *Figure 4B,C*). Informative frequencies were concentrated around 663 Hz and 165 Hz, indicating that information content about tone stimuli comes mostly from pyramidal cell activity.

Having shown that the model generalizes across different brain areas, we wanted to further investigate if it generalizes across different recording techniques. Therefore, in the third set of experiments, we acquired two-photon calcium fluorescence data from mouse CA1 while the head-fixed animal explored a 230 cm virtual track. Raw data was preprocessed to generate denoised activity traces for putative cells (n=685 regions of interest), these were then decomposed to a frequency representation using the same wavelet approach as before – only frequency bands between 0 Hz and 15 Hz being used because of the lower data rate (30 Hz) (see Materials and methods, *Figure 4E*). As before, wavelet coefficients were provided to the network as input and the only change was an increase in the number of down-sampling steps to account for the large number of ROIs (685 ROIs vs 128 channels for CA1 recordings). The network was able to accurately decode the animal's position on the track (mean error: 15.87 ± 16.33 cm, $R^2$-scores of 0.90 ± 0.03 vs. chance model −0.05 ± 0.127, *Figure 4F,G*). Using the same shuffling technique as before, we generated influence plots indicating the relative information provided by putative cells (*Figure 4E*) and frequencies (*Figure 4H*). In the frequency domain, the most informative bands were 0.33 Hz and 0.46 Hz, unsurprisingly mirroring the 1–2 s decay time of GCaMP6s (*Chen et al., 2013*). Interestingly the relative information content of individual cells was highly heterogeneous, a small subset (18.2%) of cells accounted for half (50%) of the influence – these units being distributed across the field of view with no discernible pattern (*Figure 4E*).

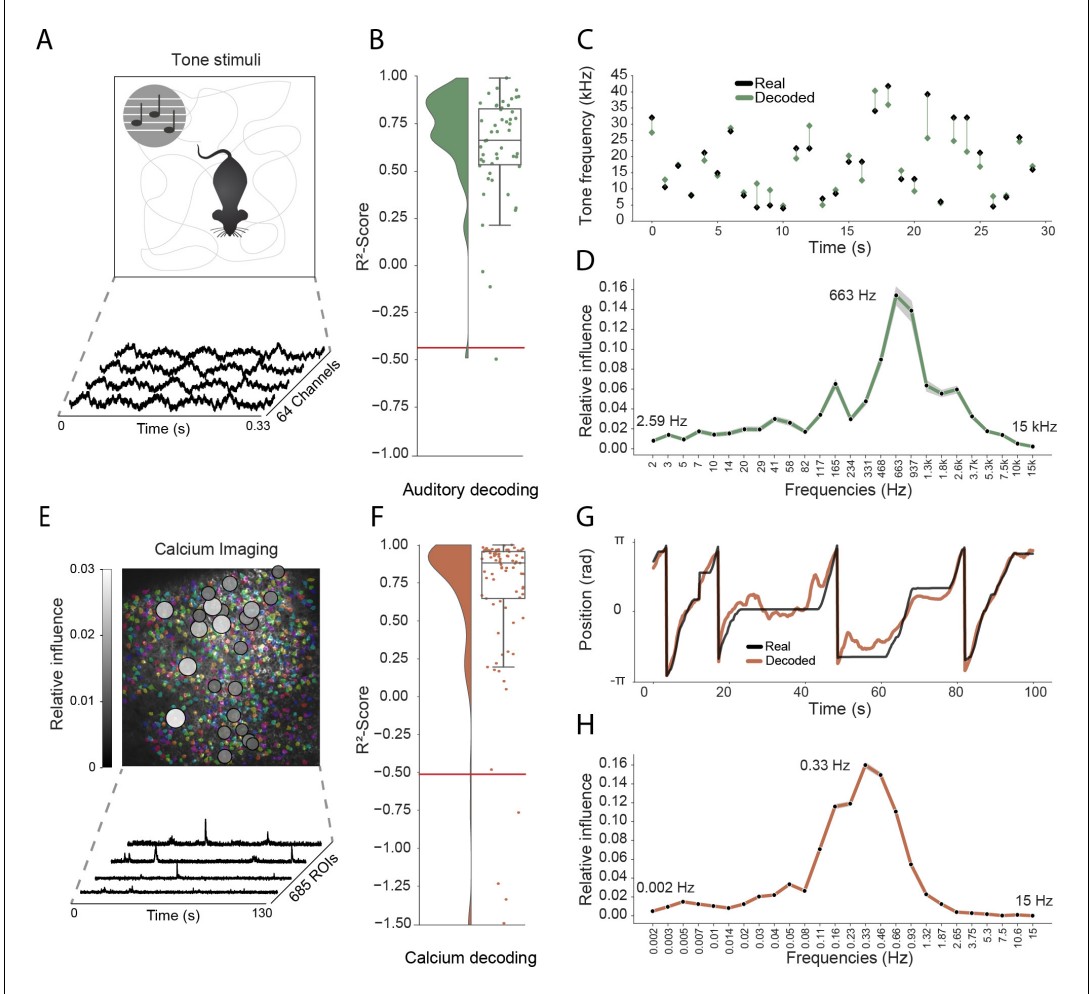

**Figure 4.** Model generalizes across recording techniques and brain regions. (**A**) Overview of auditory recording. We recorded electrophysiological signals while the mouse is freely moving inside a small enclosure and is presented with pure tone stimuli ranging from 4 kHz to 64 kHz. (**B**) $R^2$-score for decoding of frequency tone from auditory cortex ($0.73 \pm 0.08$). Each dot describes the $R^2$-score for a 5 s sample of the experiment. Chance level is indicated by the red line. (**C**) An example section for decoding of auditory tone frequencies from auditory electrophysiological recordings, real tone colored in black, decoded tone in green, the line between real and decoded indicates magnitude of error. (**D**) Influence plots for decoding of auditory tone stimuli, same method as used for CA1 recordings. (**E**) Calcium recordings from a mouse running on a linear track in VR. We record from 685 cells and use Suite2p to preprocess the raw images and extract calcium traces which we feed through the model to decode linear position. Overlay shows relative influence for decoding of position calculated for each putative cell. (**F**) $R^2$-score for decoding of linear position from two-photon CA1 recordings ($0.90 \pm 0.03$). Each dot describes the $R^2$-score for a 5 s sample of the experiment. Chance level is indicated by the red line. (**G**) Example trajectory through the virtual linear track (linearized to $[-\pi, \pi]$ with real position (black) and decoded position (orange)). (**H**) Influence plots for decoding of position from two-photon calcium imaging. Note that the range of frequencies is between 0 Hz and 15 Hz as the sampling rate of the calcium traces is 30 Hz.

The online version of this article includes the following figure supplement(s) for figure 4:

**Figure supplement 1.** Decoding finger movements from human ECoG recordings.

To further assess the ability of our model to decode continuous behavior from neural data, we investigated its performance on an Electrocorticography (ECoG) dataset recorded in humans (*Schalk et al., 2007*) made available as part of a BCI competition (BCI Competition IV, Dataset 4). In this dataset, three participants were instructed to move their fingers while ECoG signals were acquired. We used the available training data from three subjects to train and validate our model and report performance on the test set provided by the competition hosts. We used the same model pipeline, adjusting parameters to fit the specifics of the dataset. In particular, we used a downsampling factor of 50, as the original sampling rate is 1000 Hz (in comparison to 30000 Hz in

the CA1 recordings) which leads to an effective sampling rate of 20 Hz. We trained the model with 128 timesteps (6.4 s) and used a mean squared error loss function between original finger movement and decoded finger movement. All other model parameters were kept the same. We reach an average Pearson's r of 0.517 ± 0.160 across three subjects (*Figure 4—figure supplement 1A*), outperforming the winning performance of the BCI competition which reached a performance of 0.46 (note that in line with the evaluation criteria of the BCI competition we excluded the ring finger for the comparison of correlation coefficients). This allows the model to accurately predict movement of the finger, on a never before seen test set, across three different subjects (*Figure 4—figure supplement 1B*).

These results together show that our model can be used on a wide variety of continuous regression problems in both rodents and humans and across a wide range of recording systems, including calcium imaging and electrocorticography datasets.

## Discussion

The neural code provides a complex, non-linear representation of stimuli, behaviors, and cognitive states. Reading this code is one of the primary goals of neuroscience – promising to provide insights into the computations performed by neural circuits. However, decoding is a non-trivial problem, requiring strong prior knowledge about the variables encoded and, crucially, the form in which they are represented. Not only is this information often incomplete or absent but a full characterization of the neural code is precisely the question we seek to solve. Addressing these limitations, we investigated the potential of a deep-learning framework to decode behaviors and stimuli from wide-band, minimally processed neural activity. To this end, we designed a model architecture using simple 2D convolutions with shared weights, omitting recurrent layers (*Bai et al., 2018*). These intentional design choices resulted in a fast, data efficient architecture that could be easily interpreted to discover which elements of the neural code provided information about specific variables – a decrease in network performance was accepted as a trade-off. We showed that this approach generalized well across brain regions and recording techniques capturing both spatial and temporal information in the signal, the only changes necessary to the network being adjustments to handle the number of channels in the input matrix.

In the first instance, we validated our model using the well-characterized spatial representations of rodent CA1 place cells. Decoding performance amply exceeded a Bayesian framework, as well as a standard machine learning approach that proved ineffective on the non-linear representation of self-location. Importantly, simple analyses of the trained network correctly indicated that place cell action potentials were the most informative spatial signal – confirming that this tool can deliver insights into the nature of the neural code. In a further set of experiments, we showed that the network was able to concurrently identify multiple representations of head direction and running speed, including several that were only recently reported and one - interneuron encoding of head direction – that was previously unreported. Importantly, this framework can also identify interactions between frequency components, an analysis that is intractable to conventional methods which consider features independently. Finally, we demonstrated the flexibility of this approach, applying the same network and hyper-parameters, with adjustments made only to the input and output layers, to two-photon calcium data and extracellular recordings from auditory cortex.

In sum, we believe deep-learning based frameworks such as this constitute a valuable tool for experimental neuroscientists, being able to provide a general overview as to whether a variable is encoded in time-series data and also providing detailed information about the nature of that encoding – when, where, and in what frequency bands it is present. That is not to say that this approach is a complete substitute for conventional analyses – it merely constrains the search space for variables that might be present and their plausible format. Indeed, we imagine this network might be best used as a first-pass analysis, followed by conventional approaches to determine explicitly if a variable is present – much as we did for the interneuron representation of head direction. While we tested the network with optical and electrophysiological data in rodents, as well as ECoG data in humans, it is highly likely that it will perform well with neural data acquired in most experimental settings, including fMRI, EEG, and MEG.

## Materials and methods

### Tetrode recordings from CA1

Five male Lister Hooded rats were used for this study. All procedures were approved by the UK Home Office, subject to the restrictions and provisions contained in the Animals Scientific Procedures Act of 1986. All rats (333–386 g/13–17 weeks old at implantation) were implanted with two single-screw microdrives (Axona Ltd.) targeted to the right and left CA1 (ML: 2.5 mm, AP: 3.8 mm posterior to bregma, DV: 1.6 mm from dura). Each microdrive was assembled with two 32 channel Omnetics connectors (A79026-001) and 16 tetrodes of twisted wires (either 17 µm H HL coated platinum iridium, 90% and 10% respectively, or 12.7 µm HM-L coated Stablohm 650; California Fine Wire), platinum plated to reduce impedance to below 150 kΩ at 1 kHz (NanoZ). After surgery, rats were housed individually on a 12 hr light/dark cycle and after one week of recovery rats were maintained at 90% of free-feeding weight with ad libitum access to water.

Screening was performed from one week after surgery. Electrophysiological data was acquired using Open Ephys recording system (*Siegle et al., 2017*) and a 64-channel amplifier board per drive (Intan RHD2164). Positional tracking performed using a Raspberry Pi with Camera Module V2 (synchronized to Open Ephys system) and custom software, that localized two different brightness infrared LEDs attached to amplifier boards on camera images acquired at 30 Hz. During successive recording sessions in a separate screening environment 1.4 x 1.4 m the tetrodes were gradually advanced in 62.5 $\mu$m steps until place cells were identified. During the screening session, the animals were often being trained in a spatial navigation task for projects outside the scope of this study.

The experiments were run during the animals' dark period of the L/D cycle. The recording sessions used in this study were around 40 min long, depending on the spatial sampling of the animal, in a rectangular environment of 1.75 x 1.25 m, on the second, third or fourth exposure, varying between animals. The environment floor was black vinyl flooring, it was constructed of 60 cm high boundaries (MDF) colored matt black, surrounded by black curtains on the sides and above. There was one large cue card raised above the boundary and two smaller cue cards distributed on the side of the boundary. Foraging was encouraged with 20 mg chocolate-flavored pellets (LBS Biotechnology) dropped into the environment by custom automated devices. The recordings used in this study were part of a longer session that involved foraging in multiple other different size open field environments.

Rats were anesthetized with isoflurane and given intraperitoneal injection of Euthanal (sodium pentobarbital) overdose (0.5 ml/100 g) after which they were transcardially perfused with saline, followed by a 10% Formalin solution. Brains were removed and stored in 10% Formalin and 30% Sucrose solution for 3–4 days prior to sectioning. Subsequently, 50 $\mu$m frozen coronal sections were cut using a cryostat, mounted on gelatine coated or positively charged glass slides, stained with cresyl violet and cleared with clearing agent (Histo-Clear II), before covering with DPX and coverslips. Sections were then inspected using Olympus microscope and tetrode tracks reaching into CA1 pyramidal cell layer were verified.

Putative interneurons were classified based on waveform shape, minimum firing rate across multiple environments and lack of spatial stability. Specifically, classified interneurons had waveform half-width less than 0.15 ms, maximum ratio of amplitude to trough of 0.4, minimum firing rate of 4 Hz and maximal 0.75 spatial correlation of ratemaps from first and last half of the recording in any environment (*Klausberger et al., 2003*; *Csicsvari et al., 1999*). Note that we used the spatial stability in order to differentiate interneurons from place cells or grid cells, with no influence on the directional stability of the head direction cell analysis.

### Calcium recordings from CA1

All procedures were conducted in accordance to UK Home Office regulations.

One GCaMP6f mouse (C57BL/6J-Tg(Thy1-GCaMP6f)GP5.17Dkim/J, Jacksons) was implanted with an imaging cannula (a 3 mm diameter x 1.5 mm height stainless-steel cannula with a glass coverslip at the base) over CA1 (stereotaxic coordinates: AP=-2.0, ML=-2.0 from bregma). A 3 mm craniotomy was drilled at these coordinates. The cortex was removed via aspiration to reveal the external capsule of the hippocampus. The cannula was inserted into the craniotomy and secured to the skull with dental cement. A metal head-plate was glued to the skull and secured with dental

cement. The animal was left to recover for at least 1 week after surgery before diet restriction and habituation to head-fixation commenced.

Following a period of handling and habituation, the mouse was head-fixed above a styrofoam wheel and trained to run for reward through virtual reality environments, presented on 3 LCD screens that surrounded the animal. ViRMEn software (*Aronov and Tank, 2014*) was used to design and present the animal with virtual reality linear tracks. Movement of the animal on the wheel was recorded with a rotary encoder and lead to corresponding translation through the virtual track. During the experimental phase of the training, the animal was trained to run down a 230 cm linear track and was required to lick at a reward port at a fixed, unmarked goal location within the environment in order to trigger release of a drop of condensed milk. Licks were detected by an optical lick detector, with an IR LED and sensor positioned on either side of the animal's mouth. When the animal reached the end of the linear track, a black screen appeared for 2 s and the animal was presented with the beginning of the linear track, starting a new trial.

Imaging was conducted using a two-photon microscope (resonant scanning vivoscope, Scientifica) using 16x/0.8-NA water-immersion objective (Nikon). GCaMP was excited using a Ti:sapphire laser (Mai Tai HP, Spectra-Physics), operated with an excitation wavelength of 940 nm. ScanImage software was used for data collection / to interface with the microscope hardware. Frames were acquired at a rate of 30 Hz.

The Suite2p toolbox (*Pachitariu et al., 2017*) was used to motion correct the raw imaging frames and extract regions of interest, putative cells.

## Tetrode recordings from auditory cortex

Sound-evoked neuronal responses were obtained via chronically-implanted electrodes in the right hemisphere auditory cortex of one 17-week-old male mouse (*M. musculus*, C57Bl/6, Charles River). All experimental procedures were carried out in accordance with the institutional animal welfare guidelines and a UK Home Office Project License approved under the United Kingdom Animals (Scientific Procedures) Act of 1986.

During recordings, the animal was allowed to freely move within a 11x21 cm cardboard enclosure, with one wall consisting of an acoustically transparent mesh panel to allow unobstructed sound stimulation. Acoustic stimuli were delivered via two free-field electrostatic speakers (Tucker-Davis Technologies, FL, USA) placed at ear level, 7 cm from the edge of the enclosure. Recordings were performed inside a double-walled soundproof booth (IAC Acoustics), whose interior was covered by 4 cm thick acoustic absorption foam (E-foam, UK). Pure tones were generated using MATLAB (Matlab version R2015a; MathWorks, Natwick, MA, USA), and played via a digital signal processor (RX6, Tucker Davis Technologies, FL, USA). The frequency response of the loudspeaker was ±10 dB across the frequency range used for stimulation. Pure tones of a duration of 200 ms (including 5 ms linear rise and fall times) of variable frequencies (4–64 kHz in 0.1 octave increments) were used for stimulation. The tones were presented at 65 dB SPL at the edge of the testing box. The 41 frequencies were presented pseudo-randomly, separated by a randomly varying inter-stimulus interval ranging from 500 to 510 ms, for a total of 20 repetitions.

Extracellular electrophysiological recordings were obtained using a custom chronically implanted 64-channel hyperdrive with two 32-channel Omnetics connectors (A79026-001) and 16 individually movable tetrodes (FlexDrive, *Voigts et al., 2013*). Tetrodes were made from 12.7 $\mu$m tungsten wire (99.95%, HFV insulation, California Fine Wire, USA) gold-plated to reduce impedance to 200 kΩ at 1 kHz (NanoZ, Multichannel Systems). Neuronal signals were collected and amplified using two 32-channel amplifier boards (Intan RHD 2132 headstages) and an Open Ephys recording system (*Siegle et al., 2017*) at 30 kHz.

## Data preprocessing

Raw electrophysiological traces as well as calcium traces were transformed to a frequency representation using discrete-wavelet transformation (DWT). We decided to use wavelet transformation instead of windowed Fourier transform (WFT) as we expected a wide range of dominant frequencies in our signal for which the wavelet transformation is more appropriate (*Christopher and Gilbert, 1998*). For the wavelet transformation, we used the morlet wavelet:

$$\psi_0(\eta) = \pi^{-\frac{1}{4}} * \exp(i * \omega_0 * \eta) * \exp(-\eta^2/2) \tag{1}$$

with a non-dimensional frequency constant $w_0 = 6$. The full frequency space for tetrode recordings consisted of 26 log-space frequencies with fourier frequencies of: 2.59, 3.66, 5.18, 7.32, 10.36, 14.65, 20.72, 29.3, 41.44, 58.59, 82.88, 117.19, 165.75, 234.38, 331.5, 468.75, 663, 937.5, 1326, 1875, 2652, 3750, 5304, 7500, 15,000 Hz. For calcium imaging the fourier frequencies used are: 0.002, 0.003, 0.005 0.007, 0.01, 0.014, 0.02, 0.03, 0.04, 0.058, 0.08, 0.11, 0.16, 0.23, 0.33, 0.46, 0.66, 0.93, 1.32, 1.87, 2.65, 3.75, 5.3, 7.5, 10.6, 15 Hz. We noticed that downscaling the wavelets improved our model performance, prompting us to use an additional preprocessing step which effectively decreased the sampling rate of the wavelets to $\psi_{SR_{after}} = \psi_{SR_{before}}/M$ by a factor of $M$. This can also be seen as an additional convolutional layer with a kernel size of $M$, a stride of $M$ and weights fixed to $\frac{1}{M}$. We performed a hyperparameter search for $M$ with a simplified model and found the best performing model with $M = 1000$, thus effectively decreasing our sampling rate from 30,000 to 30 (*Figure 1—figure supplement 10*).

As additional preprocessing steps, we applied channel and frequency wise normalization using a median absolute deviation (MAD) approach. We calculated the median and the corresponding median absolute deviation for each frequency and channel on the training set and normalized our inputs as follows:

$$X_{c,f} = \frac{X_{c,f} - \tilde{X}_{c,f}}{\mathrm{median}(|X_{c,f} - \tilde{X}_{c,f}|)} \tag{2}$$

where $\tilde{X}_i$ is the median of $X_i$. This approach turned out to be more robust against outliers in the signal than simple mean normalization. Additional min-max scaling did not further improve performance.

## Bayesian decoder

As a baseline model, we used a Bayesian decoder which was trained on manually sorted and clustered spikes. Given a time window $T$ and number of spikes $K = (k_1, ... k_N)$ fired by $N$ place cells, we can calculate the probability $P(K|x)$, estimating the number of spikes $K$ at location $x$:

$$P(K|x) = \prod Poisson(k_i, T\alpha_i(x)) = \prod_{i=1}^{N} \frac{(T\alpha_i(x))^{k_i}}{k_i!} \tag{3}$$

where $\alpha_i(x)$ is the firing rate of cell $i$ at position $x$. From this we can calculate the probability of the animals location given the observed spikes:

$$P(x|K) = P(x) \prod_{i=1}^{N} \alpha_i(x)^{k_i} \exp(-T \sum_{i=1}^{N} \alpha_i(x)) \tag{4}$$

where $P(x)$ is the historic position of the animal which we use to constrain $P(x|K)$ to provide a fair comparison to the convolutional decoder. The final estimate of position is based on the peak of $P(x|K)$:

$$\hat{x} = argmax_x P(x|K) \tag{5}$$

We implemented a Bayesian continuity prior using a Gaussian distribution centred around the previous decoded location $x_{t-1}$ of the animal, adjusting the standard deviation based on the speed of the animal in the previous four timesteps, as implemented in *Zhang et al., 1998*. Specifically we calculated the probability of the current location given the previous by:

$$P(x_{t-1}|x_t) = \exp(\frac{-||x_t - x_{t-1}||^2}{2\sigma_t^2}) \tag{6}$$

with the standard deviation adjusted based on the speed of the animal

$$\sigma_t = v_t * V \tag{7}$$

with constant V making the fraction dimensionless. We used the previous four timesteps (2 s) to estimate the speed of the animal and used V=1s. We only used the Bayesian decoder with continuity

prior for the comparison between Bayesian decoders (see *Figure 1—figure supplement 4B*). In all other analyses, we used the Bayesian decoder without the continuity prior.

We used the same cross-validation splits as for the convolutional model and calculated the Euclidean distance between the real and decoded position. We performed grid search on one representative rat to find the optimal parameters regarding bin size and bin length for the Bayesian decoder. The optimized Bayesian decoder uses a Gaussian smoothing kernel with sigma = 1.5, a bin size of 2 cm for binning the ratemaps, and uses a bin length of 0.5 s (see *Figure 1—figure supplement 4A* for Bayesian performance across different time windows).

## Convolutional neural network

The model takes a three-dimensional wavelet transformed signal as input and uses convolutional layers connected to a regression head to decode continuous behavior. We use a kernel size of 3 throughout the model and keep the number of filters constant at 64 for the first eight layers while sharing the weights over the channel axis, while then doubling them for each following layer. For downsampling the input, we use a stride of 2, intermixed between the time and frequency dimension (Table S1, *Figure 1B*). As regularization we apply gaussian noise $\mathcal{N}(\mu, \sigma^2)$ with $\mu = 0$ and $\sigma = 1$ to each input sample.

We extensively investigated the use of a convolutional long-short-term-memory (LSTM) after the initial convolutions, where we used backpropagation through time on the time dimension. In a simplified model, this led to a small decrease in decoding error for the model trained on position. We nevertheless decided to employ a model with only simple convolutions as one important aspect of this model is the simplicity of use for a neuroscientist. Moreover, we experimented with using a wavenet (*Oord et al., 2016*) inspired model directly on the raw electrophysiological signal but noticed that the model using the wavelet transformed input outperformed the wavenet approach by a margin of around 20 cm for the positional decoding. The wavenet inspired model was considerably slower to train and therefore a full hyperparameter search could not be performed.

Previous models contrasting recurrent vs. convolutional networks (*Bai et al., 2018*), find that convolutional layers outperform recurrent ones when trained directly on minimally processed data. The benchmarks typically used in classical sequence learning are one-dimensional, whereas we record two-dimensional raw input (time x channels) with a high sampling rate, complicating the amount of experimentation we could perform as the unprocessed data for a 2 s time window exceeds the capacity of GPU memory (30,000 x 128 time points per sample). In the related field of speech processing with sampling rates up to 48,000 Hz, the input is processed using log-mel feature banks which are computed with a 25 ms window and a 10 ms shift (*Bahdanau et al., 2016*; *William et al., 2016*; *Rohit et al., 2017*). We therefore opted for a similar approach by using downsampled wavelet transformed signals, resulting in a 33.3 ms window given a downsampling size of z=1000. Note that with further downsampling there might be a risk of losing decoding precision, with some of the behaviors coming close to the downsampled rate (e.g. head direction can be up to 40 deg/s) (*Figure 1—figure supplement 10*).

## Model training

The model takes as input a three-dimensional wavelet transformed signal corresponding to time, frequency and channels, with frequencies logarithmically scaled between 0 Hz and 15.000Hz. An optimal temporal window of T=64 (corresponding to 2.13 s) was established by hyperparameter search taking into account the tradeoff between speed of training and model error. For training the model across the full duration of the experiment, we divided the experiment into five partitions and used cross-validation for testing the model on before unseen data partitions, that is we first used partitions 2 to 5 for training and one for testing, then 1, 3, 4, and 5 for training and two for testing and so on. The last partition uses 1 to 4 for training and five for testing. Importantly, the overlap introduced by using 2 s long samples was accounted for by using gaps (2 s) between the training partitions, making sure that training and test set are fully independent of each other. We then randomly sampled inputs and outputs from the training set. Each input had corresponding outputs for the position (X, Y in cm), head direction (in radians) and speed (in cm/s). We used Adam as our learning algorithm with a learning rate of 0.0007 and stopped training after we sampled 18,000 samples, divided into 150 batches for 15 epochs, each batch consisting of eight samples. During training, we

multiplied the learning rate by 0.2 if validation performance did not improve for three epochs. We performed random hyperparameter search for the following parameters: learning rate, dropout, number of units in the fully connected layer and number of input timesteps. For calculating the chance level, we used a shuffling procedure in which the wavelet transformed electrophysiological signal is shifted relative to its corresponding position. After shuffling, we trained the model with the same setting as the unshuffled model and for the same number of epochs. The training was performed on one GTX1060 using Keras with Tensorflow as backend.

## Model comparison

In order to compare the performance of the network against the Bayesian decoder we simulated both models in a setting with artificially reduced inputs. We used 1 to 32 tetrodes as input for both decoders, with tetrodes taken top to bottom in order of the given tetrode number. The input of run one was then comprised of tetrodes 1 to 32, while run 2 used tetrodes 1 to 31. The last run uses only the first tetrode as input to both models. We then retrained both models with the artificially reduced number of tetrodes making sure both models have the same cross-validation splits and report decoding errors as the average of each cross-validation split.

To generate a further baseline measure of performance when decoding using wavelet transformed coefficients, we trained support vector machines to decode position from wavelet transformed CA1 recordings. We used either a linear kernel or a non-linear radial-basis-function (RBF) kernel to train the model, using a regularization factor of C=100. For the non-linear RBF kernel we set gamma to the default $\frac{1}{num\_features * var(X)}$ as implemented in the sklearn framework. The SVM model was trained on the same wavelet coefficients as the convolutional neural network.

## Model evaluation

For adjusting the model weights during training we use different loss terms depending on the behavior or stimuli which we decode.

$$\mathcal{L}_{ED} = \sqrt{\sum_{i=1}^{M}(\hat{y}_i - y_i)^2} \qquad \mathcal{L}_{MAE} = \frac{1}{M}\sum_{i=1}^{M}|\hat{y}_i - y_i| \qquad \mathcal{L}_{CMAE} = \min[|\hat{y}_i - y_i|, |\hat{y}_i - y_i| + \pi, |\hat{y}_i - y_i| - \pi] \qquad (8)$$

For decoding of position from tetrode CA1 recordings we try to minimize the Euclidean loss between predicted and ground truth position ($\mathcal{L}_{ED}$). We use the mean squared error for the decoding of speed ($\mathcal{L}_{MAE}$) and the cyclical absolute error for decoding of head direction ($\mathcal{L}_{CMAE}$). For all other behaviors or stimuli we use $\mathcal{L}_{MAE}$ as the default optimizer.

We decided to use $R^2$ scores to measure model performance across different behaviors, brain areas, and recording techniques. We use the formulation of fraction of variance accounted for (FVAF) instead of the squared Pearson's correlation coefficient. Both terms are based on the fraction of the residual sum of squares and the total sum of squares:

$$R^2 = 1 - \frac{\sum_{i=1}^{M}(y_i - \alpha\hat{y}_i - \beta)^2}{\sum_{i=1}^{M}(y_i - \bar{y})} \qquad (9)$$

with $y_i$ the ground truth of sample $i$, $\hat{y}_i$ the predicted value and $\bar{y}$ the mean value. Here, Pearson's correlation coefficient tries to maximize $R^2$ by adjusting $\alpha$ and $\beta$ while FVAF uses $\alpha = 1$ and $\beta = 0$ (*Fagg et al., 2009*). This provides a more conservative measure of performance as FVAF requires that prediction and ground truth fit without scaling the predicted values. FVAF in turn has no lower bound as the prediction can be arbitrarily worse with a given scaling constant (i.e. given a ground truth value of 10, a prediction of 1000 has a lower (worse) $R^2$ score than a prediction of 100).

## Detection of replay events

To investigate if the model can detect replay events we reran the analysis using a lower downsampling factor. In addition, we used a new dataset in which animals performed a goal directed task with specific reward locations and in which we had previously detected sharp-wave ripples using standard methods. We applied the same preprocessing only adjusting the downsampling factor to 60, resulting in a sampling rate of 500 Hz instead of the previously used 30 Hz. We then trained the

model using the default 64 samples (128 ms) in order to obtain a better estimate of decoded positions during ripples. All other model parameters were kept the same. We then trained the model using full cross-validation and decoded the animal position every 2 ms, resulting in over 1 million decoded positions for the experiment which lasted ~35 min. Ripples in the raw neural data were extracted by finding regions in the band-pass filtered ripple band (150–250 Hz) that were above a certain threshold (five standard deviations above the mean). We expanded these until power was back at a lower threshold (0.5 standard deviations above the mean). We only kept ripples which were longer than 60 ms and occurred during stationary periods. To quantify putative replay events detected by the CNN we calculated three measures. First, we evaluated if the events showed a higher overall decoding error by calculating the average loss – Euclidean distance between decoded and true location. Second, we quantified the length of each event to detect if they are longer than the average decoded trajectory during immobility, matched to be the same length as ripple periods. Third, we generated a coherence measure to assess the smoothness or continuity of each trajectory, defined as the absolute distance between each point on the trajectory to a second degree polynomial fitted to the entire trajectory. To obtain a statistical score for all three, we permuted the events in time (keeping the absolute length of each) and recalculated each measure 1000 times. We only used periods where the speed distribution of the original replay events matched the shuffled replay events (stationary periods).

### Influence maps

To investigate which frequencies, channels or timepoints were informative for the respective decoding we performed a bootstrapping procedure after training the models. For each sample in time, we calculated the real decoding error $e_o$ for each behavior by using the wavelets as input. We then shuffle the wavelets for a particular frequency and re-calculate the error. We then define the influence of a given frequency or channel as the relative change: $\frac{e_s - e_o}{e_o}$ where $e_o$ is the original error and $e_s$ the shuffled error. We repeat this for the channel and time dimension to get an estimate of how much influence each channel or timepoint has on the decoding of a given behavior.

To evaluate if the influence measure accurately captures the true information content, we used simulated behaviors in which ground truth information was known. We used the preprocessed wavelet transformed data from one animal and created a simulated behavior ($y_{sb}$) using uniform random noise. Two frequency bands were then modulated by the simulated behavior using $f_{new} = f_{old} * \beta * y_{sb}$. We used β=2 for 58 Hz and β=1 for 3750 Hz. We then retrained the model using five-fold cross validation and evaluated the influence measure as previously described. We report the proportion of frequency bands that fall into the correct frequencies (i.e. the frequencies we chose to be modulated, 58 Hz and 3750 Hz).

We also tried calculating sample gradients with respect to our inputs (*Simonyan et al., 2013*). For this, we calculated the derivative $w$ by back-propagation for each sample and with respect to the inputs. In contrast to class saliency maps, we obtain a gradient estimate indicating how much each part of the input strongly drives the regression output. We calculate saliency maps for each sample cross-validated over the entire experiment. For deriving influence maps from the raw gradients we calculate the variance across the time dimension and use this as an estimate of how much influence each frequency band or channel has on the decoding. This method however introduces a lot of high-frequency noise in the gradients, possibly coming from the strides in the convolutional layers used throughout the model (*Olah et al., 2017*).

### Code availability

We provide the full code and notebooks demonstrating the usage of our method on electrophysiological and two-photon calcium imaging data. All examples can be adapted to predict one-dimensional (default) or n-dimensional behaviors or stimuli from any desired brain area. The code is available at https://github.com/CYHSM/DeepInsight.

## Additional information

### Funding

| Funder | Grant reference number | Author |
|---|---|---|
| H2020 European Research Council | ERC-CoG GEOCOG 724836 | Christian F Doeller |
| Wellcome Trust | 212281/Z/18/Z | Caswell Barry |
| Wellcome Trust | 110238/Z/15/Z | Catherine Perrodin |

The funders had no role in study design, data collection and interpretation, or the decision to submit the work for publication.

### Author contributions

Markus Frey, Data curation, Software, Formal analysis, Visualization, Methodology, Writing - original draft, Writing - review and editing; Sander Tanni, Catherine Perrodin, Alice O'Leary, Julie Lefort, Resources, Data curation; Matthias Nau, Methodology, Writing - review and editing; Jack Kelly, Conceptualization; Andrea Banino, Supervision, Methodology; Daniel Bendor, Funding acquisition; Christian F Doeller, Conceptualization, Supervision, Funding acquisition; Caswell Barry, Conceptualization, Resources, Data curation, Supervision, Methodology, Writing - original draft, Project administration, Writing - review and editing

### Author ORCIDs

Markus Frey (iD) https://orcid.org/0000-0003-0291-3391
Sander Tanni (iD) http://orcid.org/0000-0002-9275-0735
Matthias Nau (iD) http://orcid.org/0000-0003-0956-7815
Christian F Doeller (iD) https://orcid.org/0000-0003-4120-4600

### Ethics

Animal experimentation: All procedures were approved by the UK Home Office, subject to the restrictions and provisions contained in the Animals Scientific Procedures Act of 1986.

### Decision letter and Author response

Decision letter https://doi.org/10.7554/eLife.66551.sa1
Author response https://doi.org/10.7554/eLife.66551.sa2

## Additional files

### Supplementary files

• Supplementary file 1. Layer by layer architecture of the convolutional model. Note that the first layers 1–8 share the weights over the channel dimension while layers 9–15 share the weights across the time dimension. Layers 9 to 15 depict the kernel sizes and strides for the tetrode recordings with 128 channels. For recordings with different number of channels we adjust the number of downsampling layers to match the dimension of layer 15. Order of dimensions: Time, Frequency, Channels.

• Transparent reporting form

### Data availability

Raw neural recordings from CA1 in rodents using tetrode or two-photon calcium imaging, as well as auditory cortex recordings in mice are available here: https://figshare.com/collections/DeepInsight_-_Data_sharing/5486703. The ECoG dataset is part of a BCI competition (http://www.bbci.de/competition/iv/) and can be obtained from https://purl.stanford.edu/zk881ps0522 (fingerflex.zip).

The following datasets were generated:

| Author(s) | Year | Dataset title | Dataset URL | Database and Identifier |
|---|---|---|---|---|
| Tanni S, Barry C | 2021 | Tetrode recordings from CA1 | https://doi.org/10.6084/m9.figshare.14909766.v3 | figshare, 10.6084/m9.figshare.14909766.v3 |
| O'Leary A, Barry C | 2021 | Calcium traces from CA1 | https://doi.org/10.6084/m9.figshare.12687881 | figshare, 10.6084/m9.figshare.12687881 |
| Perrodin C, Bendor D | 2021 | Tetrode recordings from auditory cortex | https://doi.org/10.6084/m9.figshare.14909730.v2 | figshare, 10.6084/m9.figshare.14909730.v2 |

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
