## [Decision Letter]

**Acceptance summary:**

Frey et al. describe a convolutional neural network capable of extracting behavioral correlates from wide-band LFP recordings or even lower-frequency imaging data. The analysis program described by the authors provides a rapid "first pass" analysis using raw, unprocessed data to generate hypotheses that can be tested later with conventional in-depth analyses. This approach is of real value to the community, particularly as it becomes more commonplace for labs to acquire multi-site in vivo recordings.

**Decision letter after peer review:**

Thank you for submitting your article "Interpreting wide-band neural activity using convolutional neural networks" for consideration by *eLife*. Your article has been reviewed by 2 peer reviewers, and the evaluation has been overseen by a Reviewing Editor and Joshua Gold as the Senior Editor. The reviewers have opted to remain anonymous.

The reviewers agree that the tools and resources described in the manuscript are a substantial contribution to the field, but have raised a number of concerns which need to be addressed. The requested changes do not require any new data, but do require new analyses to be performed. Complete reviewer recommendations are appended at the end of this message, and a summary of essential revisions follows.

Essential revisions:

1) The CNN described in the manuscript needs to be better characterized in terms of its history dependence, comparison to Bayesian decoding, and ability to decode non-local representations. Head direction decoding using CNN needs to be compared with that using a Bayesian decoder.

2) The ability of the CNN to discover underlying truth needs to be characterized using simulations.

3) Contributions of low frequency activity need to be better distinguished from the low frequency components of excitatory spikes.

4) The amount of data required for accurately recovering the frequency bands contributing to decoding needs to be characterized using simulations as well as sub-sampled data. Relative contributions of using the Wavelet coefficients as inputs and the CNN to improved accuracy also need to be characterized.

*Reviewer #1 (Recommendations for the authors):*

In the current manuscript, Frey et al. describe a convolutional neural network capable of extracting behavioral correlates from wide-band LFP recordings or even lower-frequency imaging data. Other publications (referenced by the authors) have employed similar ideas previously, but to my knowledge, the current implementation is novel. In my opinion, the real value of this method, as the authors state in their final paragraph, is that it represents a rapid, "first-pass" analysis of large-scale electrophysiological recordings to quickly identify relevant neural features which can then become the focus of more in-depth analyses. As such, I think the analysis program described by the authors is of real value to the community, particularly as it becomes more commonplace for labs to acquire multi-site in vivo recordings.

However, to maximize its utility to the community, I have several questions/concerns that I believe need to be addressed.

(1) It is obviously important to quantify the relative accuracy of the authors' method to existing methods which correlate neural activity to behavior or sensory input. The authors attempt to do this by comparing CNN decoding to Bayesian decoding with clustered cells (Figure 1). However, I think there are several points where that comparison may be flawed.

(1a) First, while some manuscripts (including Zhang et al., 1998, referenced by the authors) do use a continuity prior in their decoding algorithms, most do not (see the vast array of papers from Matt Wilson, David Redish, Loren Frank, David Foster, and others in the field). Indeed, even Olafsdottir et al., 2015 that the authors reference in apparent support of the use of a continuity prior (Line 111 of the manuscript) explicitly state that they do not use a continuity prior in their methods. It is unclear to me in reading through the methods whether the CNN-based decoding also utilizes a continuity prior to restrict the decoded location.

To truly be a useful tool to the community, the algorithm should be capable of correlating neural activity to behaviors/sensory input in a manner that is history-independent, as it seems that a fundamental advantage of this system is for unbiased probing of such correlations. If such history-independent decoding is not possible with the CNN, this should be explicitly stated to clarify the parameters for which this method is appropriate.

Thus, I would like clarification of whether the CNN uses the animal's history to constrain its decoding output. If so, is the use of the animal's history a necessary component of the CNN-based decoding? If the CNN can be used in a history-independent manner, I would like to see it compared to history-independent Bayesian decoding.

(1b) The authors report that part of the advantage of the CNN over Bayesian decoding was that the CNN made fewer large errors, and that the median error was more similar across the two methods (Line 120). When performing the Bayesian decoding, it appears as though spikes throughout the entire experiment were used. However, it is known that during periods of immobility, population bursts during sharp-wave/ripples can encode virtual trajectories across the environment, producing non-local spatial representations. If such remote trajectories were included in the Bayesian decoding, this may account for large "errors" between the decoded location and the animal's actual location (even though this may not be an error at all!). Thus, I would like to see this comparison repeated using only periods of active movement, when the literature suggests that place cells are more likely to encode local spatial information.

In addition, it appears that the authors use a 500 ms window to quantify Bayesian decoding accuracy, but, from what I can tell, they use a ~33 ms window (within 2 second 'chunks') to quantify CNN decoding accuracy. This doesn't seem like a fair apples-to-apples comparison as the animal can move quite a bit over 500 ms. I would thus like to see a comparison between these two methods using similar timescales that are appropriate for both methods, perhaps a ~100-200 ms window.

(1c) Related to the previous point, a fundamental advantage of using single unit activity to examine behavioral information is that it allows the experimenter to identify when the neural population is representing information other than the immediate sensory input or behavioral output. For example, in the place cell field, one can correlate activity of single units to position during active behavior, and then study when virtual paths are encoded during hippocampal sharp-wave/ripples (a.k.a. replay) or theta sequences.

Thus, a basic question is whether such non-local representation is accessible in the authors' CNN. Can the authors train the network on behavioral data (using only periods of active movement) and then faithfully decode virtual trajectories during immobility-based ripples? Alternatively, can the authors identify the shorter theta sequences observed during active movement if the CNN operates on a finer timescale? If the position decoding is largely based on high-frequencies in the LFP representing action potentials, it seems that such finer-scale, non-local representations should also be available. Importantly, if such non-local or fine-time-scale representations cannot be identified using this method, it is important to clarify this to avoid incorrect future use.

(2) Although the majority of self-location information was present in high-frequency bands, the data in Figure 1E show that LFP frequencies below 250 Hz were also informative above chance (and very near Bayesian decoding accuracy). However, given that Figure 3B shows excitatory spikes spread well into low 200 Hz frequency ranges, it seems that using LFP below 250 Hz is likely also including some spike information. For clarification of how accurately low-frequency bands can reflect position information, I would like to see the analysis in Figure 1E performed with LFP frequencies less than 150 Hz (fast gamma and slower).

If position information can still be faithfully extracted using <150 Hz frequencies, it is important to further rule out non-spatial correlates. For example, are there spatial locations where the rat is more likely to run at predictable velocities, allowing theta-band frequencies to effectively decode the animal's location? Is the LFP-based decoding more/less accurate at specific locations of the environment (near walls, near a rewarded location, etc.)? A heat-map of average decoding error per spatial bin (per animal) would be useful to visualize this analysis.

(3) When quantifying the accuracy of head direction decoding, they compare the CNN to chance levels. Although this is a valuable measure, given that head direction seems heavily driven by LFP frequencies associated with excitatory and inhibitory spiking, can the authors also compare head direction decoding between the CNN and Bayesian decoding from clustered spikes (including both exc. and inh. cells)? Is head direction information available in the clustered data or are there other elements in the high-frequency LFP that correlate to head direction? If head direction can be decoded via clustered spikes, why do the authors think this has not been observed in prior studies?

(4) In the Methods (line 356), I'm not sure what the authors mean by "16 eight tetrodes". Do they mean 16 tetrodes?

(5) The x-axis of Figure 4D and 4H should be labeled, especially since the scale seems to be log rather than linear.

*Reviewer #3 (Recommendations for the authors):*

– I think this method could be very useful for the EEG/ECoG communities, which care about frequency representations, and appealing to those communities would significantly expand the utility of your method for the broader neuroscience community. In my opinion, if there is no example of this type of use in the paper, it is much less likely for EEG/ECoG researchers to actually use your method in practice. I think that having an EEG or ECoG example would be much more beneficial than the calcium imaging example, since researchers are not trying to determine what frequency contents are important within a calcium imaging signal. This has a list of many open datasets for EEG: https://github.com/meagmohit/EEG-Datasets

– I think providing a general overview of your approach/method at the beginning of Results would be helpful for many readers.

– Please clarify when you are reporting test-set vs. training set predictions in your results.

– In the final paragraph of the introduction, you write "Our model differs markedly from conventional decoding methods which typically use Bayesian estimators…" This is overly specific to hippocampal decoding – in movement decoding Bayesian methods are not frequently used, although linear methods still commonly are.

---

## [Author Response]

Reviewer #1 (Recommendations for the authors):In the current manuscript, Frey et al. describe a convolutional neural network capable of extracting behavioral correlates from wide-band LFP recordings or even lower-frequency imaging data. Other publications (referenced by the authors) have employed similar ideas previously, but to my knowledge, the current implementation is novel. In my opinion, the real value of this method, as the authors state in their final paragraph, is that it represents a rapid, "first-pass" analysis of large-scale electrophysiological recordings to quickly identify relevant neural features which can then become the focus of more in-depth analyses. As such, I think the analysis program described by the authors is of real value to the community, particularly as it becomes more commonplace for labs to acquire multi-site in vivo recordings.However, to maximize its utility to the community, I have several questions/concerns that I believe need to be addressed.(1) It is obviously important to quantify the relative accuracy of the authors' method to existing methods which correlate neural activity to behavior or sensory input. The authors attempt to do this by comparing CNN decoding to Bayesian decoding with clustered cells (Figure 1). However, I think there are several points where that comparison may be flawed.(1a) First, while some manuscripts (including Zhang et al., 1998, referenced by the authors) do use a continuity prior in their decoding algorithms, most do not (see the vast array of papers from Matt Wilson, David Redish, Loren Frank, David Foster, and others in the field). Indeed, even Olafsdottir et al., 2015 that the authors reference in apparent support of the use of a continuity prior (Line 111 of the manuscript) explicitly state that they do not use a continuity prior in their methods. It is unclear to me in reading through the methods whether the CNN-based decoding also utilizes a continuity prior to restrict the decoded location.To truly be a useful tool to the community, the algorithm should be capable of correlating neural activity to behaviors/sensory input in a manner that is history-independent, as it seems that a fundamental advantage of this system is for unbiased probing of such correlations. If such history-independent decoding is not possible with the CNN, this should be explicitly stated to clarify the parameters for which this method is appropriate.Thus, I would like clarification of whether the CNN uses the animal's history to constrain its decoding output. If so, is the use of the animal's history a necessary component of the CNN-based decoding? If the CNN can be used in a history-independent manner, I would like to see it compared to history-independent Bayesian decoding.

We thank the reviewer for raising this important point about the decoding details of the Bayesian decoder and the history dependence of our model. Indeed, as stated by the reviewer, most prior work does not use a continuity prior for Bayesian decoding. The reviewer is also correct in spotting that we made a mistake in referring to our Bayesian model as a decoder with a continuity prior when in fact we did not use such a prior. We apologise for this mistake which we have now corrected in the text.

In addition we now add a further analysis comparing the CNN with Bayesian decoders with and without a continuity prior. Specifically, we implemented the continuity prior by using a Gaussian distribution centred around the previous decoded location (t-1) of the animal, adjusting the standard deviation based on the speed of the animal in the previous timesteps (last 2 seconds). As shown in Figure 1 – Supplement 4B, the difference between the median values of the Bayesian decoder with and without continuity prior is small (median decoding error with continuity, 22.51cm; without continuity, 23.23cm), although the Bayesian decoder with continuity shows higher variance and makes more ‘catastrophic’ errors which is reflected by a higher mean decoding error (mean decoding error with continuity, 33.06cm; without continuity, 23.38cm). Regardless, in both cases, the CNN yields more accurate decoding (mean decoding error of 17.31cm, Figure 1).

With regards to the history dependence of our model, the CNN is feedforward and only has access to information from across the input window, which has a size of 2.13 seconds (64 timesteps). There is no explicit continuity prior as implemented in the Bayesian decoder, however during training, the network weights will learn an implicit prior based on the behaviour of the animal in the training data. This will likely incorporate information about the statistics of the animal’s motion (e.g. distance travelled between neighbouring timesteps).

We also ran an analysis across all time windows for both the Bayesian decoder as well as the convolutional neural network (see also Question 3). As seen in Figure 1 Supplement 4A, for very small time windows, which imply a history independent manner of decoding behaviour, both models show decreased performance in comparison to longer time windows.

Page 4:

“To provide a familiar benchmark, we applied a standard Bayesian decoder without a continuity prior (Olafsdottir et al. 2015) to the spiking data from the same datasets (see methods, see also Figure 1 – Supplement 4B for comparison to Bayesian decoder with continuity prior (Zhang et al. 1998)).”

Page 17:

“We implemented a Bayesian continuity prior using a Gaussian distribution centred around the previous decoded location x_t-1_ of the animal, adjusting the standard deviation based on the speed of the animal in the previous 4 timesteps, as implemented in (Zhang et al. 1998). […] We used the previous 4 timesteps (2 seconds) to estimate the speed of the animal and used V=1s.”

(1b) The authors report that part of the advantage of the CNN over Bayesian decoding was that the CNN made fewer large errors, and that the median error was more similar across the two methods (Line 120). When performing the Bayesian decoding, it appears as though spikes throughout the entire experiment were used. However, it is known that during periods of immobility, population bursts during sharp-wave/ripples can encode virtual trajectories across the environment, producing non-local spatial representations. If such remote trajectories were included in the Bayesian decoding, this may account for large "errors" between the decoded location and the animal's actual location (even though this may not be an error at all!). Thus, I would like to see this comparison repeated using only periods of active movement, when the literature suggests that place cells are more likely to encode local spatial information.

We thank the reviewer for the suggestion to compare decoding performance between stationary periods and periods of active movement.

For all analysis reported currently in the manuscript the Bayesian decoder is only applied to periods when animals are travelling >3cm/s. No speed threshold was applied to the convolutional neural network. We have now clarified this in the text. In addition, we now include a new Supplementary Figure which shows the comparison between both models across a range of speed thresholds. As expected, the performance of both models increases as the speed threshold is increased, presumably because – as the reviewer suggested – non-locomotor neural activity is excluded. The mean decoding performance of our model improves from 17.37cm ± 3.58cm with no speed threshold to 13.40cm ± 3.59cm with a speed threshold of 25cm/s, while the Bayesian decoder improves from 22.23cm ± 3.72 to 17.17cm ± 2.78.

Importantly the CNN is more accurate than the Bayesian decoder for all thresholds.

Page 4:

“Note that for these comparisons the Bayesian decoder was only applied to periods when the animal was travelling at >3cm/s, in contrast the CNN did not have a speed threshold (i.e. is trained on moving and stationary periods). […] As expected, limiting the CNN to only periods of movement actually improves its performance, accentuating the difference between it and the Bayes decoder (0cm/s speed threshold: Bayesian mean error 22.23cm ± 3.72cm; network error 17.37cm ± 3.58 cm; 25cm/s speed threshold: Bayesian mean error 17.17cm ± 2.788cm; network error 13.40cm ± 3.59 cm, Figure 1 Supplement 6).”

In addition, it appears that the authors use a 500 ms window to quantify Bayesian decoding accuracy, but, from what I can tell, they use a ~33 ms window (within 2 second 'chunks') to quantify CNN decoding accuracy. This doesn't seem like a fair apples-to-apples comparison as the animal can move quite a bit over 500 ms. I would thus like to see a comparison between these two methods using similar timescales that are appropriate for both methods, perhaps a ~100-200 ms window.

We thank the reviewer for this question regarding the timeframes of both the Bayesian decoder and our model. Indeed the Bayesian decoder uses a 500ms window to decode position in the environment, while our model uses a 2s window, from which 4 separate timesteps are decoded (every 500ms). However, we only analyse the last of the decoded sample – ensuring that the model is not able to decode based on future neural data. In effect, all reported performance scores for our model use a 2s window, in which the last timestep is decoded.

We now include an additional Supplementary Figure comparing the performance of both the Bayesian decoder and the convolutional neural network across different durations, ranging from 100ms up to 1.6 seconds. Both models yield more accurate decoding with longer time windows – which incorporate more data – Bayesian being most accurate with a 1s window, the CNN performing best with a 1.6s window (Figure 1 – Supplement 4A). Here again, CNN performance is better than the Bayes decoder for all time windows.

Page 17:

“The optimized Bayesian decoder uses a Gaussian smoothing kernel with σ = 1.5, a bin size of 2cm for binning the ratemaps, and uses a bin length of 0.5s (see Figure 1 Supplement 4A for Bayesian performance across different time windows).”

(1c) Related to the previous point, a fundamental advantage of using single unit activity to examine behavioral information is that it allows the experimenter to identify when the neural population is representing information other than the immediate sensory input or behavioral output. For example, in the place cell field, one can correlate activity of single units to position during active behavior, and then study when virtual paths are encoded during hippocampal sharp-wave/ripples (a.k.a. replay) or theta sequences.Thus, a basic question is whether such non-local representation is accessible in the authors' CNN. Can the authors train the network on behavioral data (using only periods of active movement) and then faithfully decode virtual trajectories during immobility-based ripples? Alternatively, can the authors identify the shorter theta sequences observed during active movement if the CNN operates on a finer timescale? If the position decoding is largely based on high-frequencies in the LFP representing action potentials, it seems that such finer-scale, non-local representations should also be available. Importantly, if such non-local or fine-time-scale representations cannot be identified using this method, it is important to clarify this to avoid incorrect future use.

This is a really interesting suggestion – we thank the reviewer for making it.

In the original implementation of the convolutional model we downsample the electrophysiological signals after the wavelet transformation by a factor of 1000 to maximize the size (in seconds) of the input sample – the amount of data in a sample is constrained by GPU memory size. It’s highly likely this preprocessing would compress short replay events, making them hard to decode.

Thus to investigate if the model can detect replay events we reran the analysis using less downsampling (see Figure 1—figure supplement 7). In addition we used a new dataset in which rats were performing a navigation task and for which we had observed replay events before. We used the same preprocessing, only adjusting the downsampling factor to 60, resulting in a sampling rate of 500 Hz instead of the previously used 30 Hz. We then trained the model using 64 samples (128ms) in order to obtain a better estimate of decoded positions during replay events. We then decoded the animal position every 2 ms, resulting in over 1 million decoded positions for the experiment which lasted ~35 minutes.

To quantify if our model can detect replay we investigated model behaviour at time points where sharp-wave ripples (SWRs) had been detected using standard methods (i.e. finding LFP segments in which ripple band power (150-250Hz) was at least 5 standard deviations above the mean and expanding these regions until power dropped back to 0.5 standard deviations above the mean, only retaining segments that were longer than 60ms). Examining these SWRs we saw the CNN often decoded transient, high velocity trajectories that resembled those reported in studies of open field replay (see Figure 1 – Supplement 7A). Consistent with this interpretation the decoding error for these periods (i.e. Euclidean distance between the animal’s location and decoded location) was larger than for matched periods when the animal was stationary but in which SWRs had not been detected (n=1000, p<0.001, Figure 1 Supplement 7B). We furthermore found these putative replay trajectories were longer than the average decoded trajectories during stopped periods (n=1000, p=0.003, Figure 1 – Supplement 7C) and they were as coherent as trajectories detected during movement (n=1000, p=0.257, Figure 1 – Supplement 7D).

Page 6:

“The standard decoding model downsamples the wavelet frequencies to a rate of 30Hz, potentially discarding transient non-local representations (e.g. replay events and theta sequences). […] Thus it seems plausible that non-local representations are accessible to this CNN framework.”

Page 19-20:

“To investigate if the model can detect replay events we reran the analysis using a lower downsampling factor. […] We only used periods where the speed distribution of the original replay events matched the shuffled replay events (stationary periods).”

(2) Although the majority of self-location information was present in high-frequency bands, the data in Figure 1E show that LFP frequencies below 250 Hz were also informative above chance (and very near Bayesian decoding accuracy). However, given that Figure 3B shows excitatory spikes spread well into low 200 Hz frequency ranges, it seems that using LFP below 250 Hz is likely also including some spike information. For clarification of how accurately low-frequency bands can reflect position information, I would like to see the analysis in Figure 1E performed with LFP frequencies less than 150 Hz (fast gamma and slower).If position information can still be faithfully extracted using <150 Hz frequencies, it is important to further rule out non-spatial correlates. For example, are there spatial locations where the rat is more likely to run at predictable velocities, allowing theta-band frequencies to effectively decode the animal's location? Is the LFP-based decoding more/less accurate at specific locations of the environment (near walls, near a rewarded location, etc.)? A heat-map of average decoding error per spatial bin (per animal) would be useful to visualize this analysis.

This was a good suggestion, it does indeed appear that the reviewer was correct – the majority of the spatial information is contained in the 125 to 250 Hz band.

We reran our LFP decoding analysis the same way as before but further segregated LFP frequencies into two bands – 0-150Hz (12 of 26 frequencies) and 150-250Hz (3 of 26 frequencies). As shown in Figure 1 – Supplement 2, spatial decoding performance was much lower when only frequencies up to 150 Hz were used, relative to the model evaluated on all frequencies up to 250 Hz. In contrast, the models trained on the 150-250Hz band and 0-250Hz band performed similarly and were sufficient to reach the same performance as the Bayesian decoder (shown in grey).

Indeed, as can be seen in Figure 3 – Supplement 1, models which were only trained on one frequency band at a time demonstrated accurate spatial decoding on frequencies from 165Hz upwards This together with the previous analysis support the reviewer’s hypothesis that excitatory spikes are picked up in frequency bands lower than the traditional 250Hz LFP cut-off. We have now revised our statements in the manuscript and point toward Figure 1 – Supplement 2 showing the difference in decoding performance for all low-frequency models.

Page 4:

“The high accuracy and efficiency of the model for these harder samples suggest that the CNN utilizes additional information from sub-threshold spikes and those that were not successfully clustered, as well as nonlinear information which is not available to the Bayesian decoder.”

Page 5: [Figure caption]

“When only local frequencies were used (<250Hz, CNN-LFP), network performance dropped to the level of the Bayesian decoder (distributions show the five-fold cross validated performance across each of five animals, n=25). Note that this likely reflects excitatory spikes being picked up at frequencies between 150 and 250 Hz (Figure 1 Supplement 2).”

Page 6:

“If we retrain the model on frequency bands 0-150 Hz and 150-250 Hz we observe that spatial information is predominantly contained in higher band frequencies (Figure 1 Supplement 2, see also Figure 3 – Supplement 1), likely reflecting power from pyramidal cell waveforms reaching these frequencies.”

(3) When quantifying the accuracy of head direction decoding, they compare the CNN to chance levels. Although this is a valuable measure, given that head direction seems heavily driven by LFP frequencies associated with excitatory and inhibitory spiking, can the authors also compare head direction decoding between the CNN and Bayesian decoding from clustered spikes (including both exc. and inh. cells)? Is head direction information available in the clustered data or are there other elements in the high-frequency LFP that correlate to head direction? If head direction can be decoded via clustered spikes, why do the authors think this has not been observed in prior studies?

We thank the reviewer for the suggestion to run our head direction decoding also with a Bayesian decoder. We now include an additional analysis in which we quantify head direction decoding using a Bayesian decoding framework, where we calculate errors based on a circular loss, similar to the way the CNN is decoding head direction. The Bayesian decoder's accuracy across all 5 rats is 55.73 ± 8.07 degrees. In comparison, our model achieves a decoding accuracy of 34.37 ± 6.87 degrees. Thus both methods are significantly better than chance (Bayesian model, Wilcoxon signed-rank test two-sided (n=25): T=0, p=1.22e-5, CNN Wilcoxon signed-rank test two-sided (n=25): T=0, p=1.22e-5), although the CNN provides a significant benefit beyond simple Bayesian decoding (Wilcoxon signed-rank test two-sided (n=25): T=47, p=0.0018). It is not entirely surprising that it is possible to decode head direction from CA1 neurons, it has been known for some time that place cells are weakly modulated by head direction (Muller et al. 1994) and, as we report in this manuscript, hippocampal interneurons are similarly modulated. However, in the case of place cells at least, the non-linear interaction between direction and position codes likely impacts the accuracy of the Bayes decoder but is less of a problem for the CNN.

Page 7:

“Note that, a Bayesian decoder trained to decode head direction achieves a performance of 0.97rad ± 0.14rad using spike sorted neural data, significantly worse than our model (Wilcoxon signed-rank test two-sided (n=25): T=47, p=0.0018) but more accurate than would be expected by chance (Wilcoxon signed-rank test two-sided (n=25): T=0, p=1.22e-5).”

(4) In the Methods (line 356), I'm not sure what the authors mean by "16 eight tetrodes". Do they mean 16 tetrodes?

We thank the reviewer for spotting this mistake which we have now revised. We indeed use 16 tetrodes per microdrive, resulting in 128 channels (16 tetrodes x 4 channels per tetrode x 2 microdrives).

(5) The x-axis of Figure 4D and 4H should be labeled, especially since the scale seems to be log rather than linear.

Indeed the frequencies are log scaled and we initially decided to only show some frequencies to not clutter the axis. We now changed the figure axis to show every frequency component. For completeness we now also report all frequencies in the methods section.

Page 16:

“The full frequency space for tetrode recordings consisted of 26 log-space frequencies with Fourier frequencies of: 2.59, 3.66, 5.18, 7.32, 10.36, 14.65, 20.72, 29.3, 41.44, 58.59, 82.88, 117.19, 165.75, 234.38, 331.5, 468.75, 663, 937.5, 1326, 1875, 2652, 3750,

5304, 7500, 15000 Hz. For calcium imaging the Fourier frequencies used are: 0.002,

0.003, 0.005 0.007, 0.01, 0.014, 0.02, 0.03, 0.04, 0.058, 0.08, 0.11, 0.16, 0.23, 0.33, 0.46, 0.66, 0.93, 1.32, 1.87, 2.65, 3.75, 5.3, 7.5, 10.6, 15 Hz.”

Reviewer #3 (Recommendations for the authors):– I think this method could be very useful for the EEG/ECoG communities, which care about frequency representations, and appealing to those communities would significantly expand the utility of your method for the broader neuroscience community. In my opinion, if there is no example of this type of use in the paper, it is much less likely for EEG/ECoG researchers to actually use your method in practice. I think that having an EEG or ECoG example would be much more beneficial than the calcium imaging example, since researchers are not trying to determine what frequency contents are important within a calcium imaging signal. This has a list of many open datasets for EEG: https://github.com/meagmohit/EEG-Datasets

We thank the reviewer for the suggestion to evaluate our model on another dataset from a different neuroscience community to increase the potential impact of our framework.

We first want to point out that frequency contents might also be important for calcium imaging researchers. One or two-photon microscopes are now being developed for freely moving animals and the decrease in size, which is necessary to fit the animal, is most often accompanied by a decrease in sampling rate. Our decoding results and the underlying informative frequencies show that a sampling rate of around 1Hz is enough to capture most of the information contained in the signal. This information can give other researchers a lower bound for the sampling rate of the neural signal.

As suggested, we now evaluate our model on a publicly available ECoG dataset (Schalk et al. 2007), in which participants finger movements were recorded while simultaneously acquiring ECoG signals. As this dataset was part of a BCI competition (BBCI IV) we can directly compare our model results to the best models of the competition. We used the available training data from three subjects to train and validate our model and report performance on the provided test set. Each subject was instructed to move a given finger in response to a visual cue, which lasted around 2 seconds. We used the same model pipeline, adjusting parameters to fit the dataset. In particular, we used a downsampling factor of 50, as the original sampling rate is 1000Hz (in comparison to 30000 Hz in the CA1 recordings) which leads to an effective sampling rate of 20Hz. We trained the model with 128 timesteps (6.4s) and used a mean squared error loss function between original finger movement and decoded finger movement. As can be seen in Figure 4 – Supplement 1 we reach an average Pearson's r of 0.517 +- 0.160 across three subjects. The best result in the competition reached a performance of 0.46 (see competition results).

Page 12:

“To further assess the ability of our model to decode continuous behaviour from neural data, we investigated its performance on an Electrocorticography (ECoG) dataset recorded in humans (Schalk et al. 2017) made available as part of a BCI competition (BCI Competition IV, Dataset 4). […] These results together show that our model can be used on a wide variety of continuous regression problems in both rodents and humans and across a wide range of recording systems, including calcium imaging and electrocorticography datasets.”

– I think providing a general overview of your approach/method at the beginning of Results would be helpful for many readers.

We thank the reviewers for this suggestion. We have now added a paragraph explaining how the Results section is structured and provide a general overview of our approach in more detail.

Page 3:

“In the following section, we present our model, the results, and describe how it was applied across different datasets. […] The transformed data is then aligned with one or several decoding outputs, representing different behaviours or stimuli, which are fed through a convolutional neural network, decoding each output separately.”

– Please clarify when you are reporting test-set vs. training set predictions in your results.

We are sorry for the lack of clarity regarding the use of training vs. test-set. We now clarify in the manuscript that all of the reported predictions are fully cross-validated on the test set and at no point in the manuscript do we report training predictions except when explicitly stated (e.g. Figure S1).

Page 3:

“Using the wavelet coefficients as inputs, the model was trained in a supervised fashion using error backpropagation with the X and Y coordinates of the animal as regression targets. We report test-set performance for fully cross-validated models using 5 splits across the whole duration of the experiment.”

– In the final paragraph of the introduction, you write "Our model differs markedly from conventional decoding methods which typically use Bayesian estimators…" This is overly specific to hippocampal decoding – in movement decoding Bayesian methods are not frequently used, although linear methods still commonly are.

We thank the reviewer for the suggestion to open up our introduction to a wider audience. We now rewrote the manuscript in the following way.

Page 2:

“Our model differs markedly from conventional decoding methods which often use Bayesian estimators (Zhang et al. 1998) for hippocampal recordings in conjunction with highly processed neural data or linear methods for movement decoding from EEG or

ECoG signals (Antelis et al. 2013).”